# Collaboration with Dynamic Open Ad Hoc Team via Team State Modelling

**Jing Sun**  *krystalsun@cityu.edu.mo*
*Faculty of Data Science*
*City University of Macau*

**Cong Zhang**[*]  *cong.zhang92@gmail.com*
*College of Computing and Data Science*
*Nanyang Technological University*

**Zhiguang Cao**  *zhiguangcao@outlook.com*
*School of Computing and Information Systems*
*Singapore Management University*

**Reviewed on OpenReview:** *https://openreview.net/forum?id=BukMU42P3G*

## Abstract

Open ad hoc teamwork requires autonomous agents to achieve rapid, adaptive collaboration with teammates without prior coordination in an open environment. Existing methods primarily rely on fixed, predefined teammate types, overlooking the fact that teammates may change dynamically. To address this limitation, we propose a novel reinforcement learning approach, the Open Online Teammate Adaptation Framework (Open-OTAF), which enables a controlled agent to collaborate with dynamic teammates in open ad hoc environments. To achieve this, the controlled agent employs a dual teamwork situation inference model to capture the current teamwork state, facilitating decision-making under partial observability. To handle the dynamic nature of teammate types, we first introduce a Chinese Restaurant Process-based model to categorize diverse teammate policies into distinct clusters, improving the efficiency of identifying teamwork situations. Next, to model heterogeneous agent relationships and accommodate a variable number of teammates, we represent the team as a heterogeneous graph and leverage heterogeneous graph attention neural networks to learn the representation of the teamwork situation. Extensive experiments across four challenging multi-agent benchmark tasks—Level-Based Foraging, Wolf-Pack, Cooperative Navigation, and FortAttack—demonstrate that our method successfully enables dynamic teamwork in open ad hoc settings. Open-OTAF outperforms state-of-the-art methods, achieving superior performance with faster convergence. These results validate Open-OTAF's capacity to adapt to unknown teammates while maintaining computational efficiency.

## 1 Introduction

Recently, multi-agent reinforcement learning (MARL) has garnered growing interest for addressing complex multi-agent tasks, such as navigation in human-shared environments and real-time strategy games (Roesch et al., 2020; Nguyen et al., 2020; Boldrer et al., 2022; Zhou et al., 2021; Zhang et al., 2018). A prominent framework in this domain is centralized training with decentralized execution (CTDE), which enables agents to train with global information in a centralized manner while making decisions based only on local observations during execution in a decentralized manner. Numerous MARL methods have been developed within the CTDE framework, primarily categorized into policy-based and value-based approaches (Lowe

---

[*]Corresponding Author.

et al., 2017; Rashid et al., 2018; Sunehag et al., 2017; Yu et al., 2022). Policy-based methods, such as MAD-DPG (Lowe et al., 2017), COMA (Foerster et al., 2018), and MAPPO (Yu et al., 2022), leverage observations and actions of controlled agents to train policy networks. In contrast, value-based methods, exemplified by VDN (Sunehag et al., 2017) and QMIX (Rashid et al., 2018), construct a joint value function to optimize team performance. However, these MARL approaches assume a fixed team configuration, where factors such as team size, formation, and goals remain unchanged, and teammate types are known and static.

In many real-world applications, agents must collaborate with unknown and diverse teammates in real-time, giving rise to the ad hoc teamwork (AHT) problem (Stone et al., 2010). Existing research on AHT often assumes that all teammates' behaviours are predefined out of types, each corresponding to a specific coordination strategy (Albrecht & Stone, 2018; Barrett et al., 2017). However, in dynamic environments, teammates' strategies may evolve unpredictably, leading to severe miscoordination if assumed to be static and fixed (Chen et al., 2020). A common approach to adapt to dynamic teammates is to leverage Bayesian posteriors of teammate types and incorporate them into reinforcement learning when learning policies (Barrett et al., 2017; Ravula, 2019; Chen et al., 2020). These methods require access to teammates' observations and actions to compute the posteriors, which is infeasible in partially observable environments. Recent efforts have attempted to address partial observability, but they primarily focus on closed environments where the number of agents remains fixed (Gu et al., 2021; Ribeiro et al., 2022; Zhang et al., 2023; Rahman et al., 2024; Li et al., 2024; Chen et al., 2025). However, many real-world scenarios require adaptation to a dynamically changing number of agents. For example, an autonomous vehicle must adjust its driving behaviour based on the number of surrounding vehicles, which could be driven by humans or produced by different manufacturers, each with distinct driving styles (Barrett & Stone, 2015; Albrecht et al., 2021). Achieving online adaptation with dynamic teammates in open AHT remains an open challenge, especially under partial observability where agents must infer teammate behaviors and intentions from limited observations. While communication and message-passing methods (Foerster et al., 2016; Das et al., 2019) have shown promise in improving belief-sharing and coordination, they often introduce additional computational costs, especially when agents cooperate with dynamic teammates (Albrecht & Ramamoorthy, 2016; Rahman et al., 2021a). This trade-off between policy adaptability and computational scalability highlights the critical need for efficient collaboration frameworks in open environment.

To address the issues mentioned above, we propose the Open Online Teammate Adaptation Framework (Open-OTAF), a novel reinforcement learning framework that explicitly models the teamwork situations, which indicates the influence on the environmental dynamics caused by other teammates, to enhance coordination in open environments. Concretely, Open-OTAF employs a dual-structured model to learn a fixed-size teamwork situation embedding, capturing both agent-specific and team-level coordination patterns. To avoid redundant encodings of similar teammates, we leverage a Chinese Restaurant Process (CRP) (Blei & Frazier, 2011) to probabilistically cluster teammates by behavioral similarity, generating an ad hoc belief over team composition. This clustering reduces the context search space and improves representation efficiency. To address dynamic team compositions and heterogeneous interactions, we represent the team as a heterogeneous graph and process it through a Heterogeneous Graph Attention Network (HAN) to generate role-aware and relationally structured teamwork situation embeddings. For policy training, we sample representative teammates during centralized training, encoding their identities into distinct contexts to enrich joint policy learning. During decentralized execution, agents approximate the global context using local observations. This iterative process yields a robust policy capable of adapting to unseen teammates and dynamic team configurations. To summarize, we highlight the following contributions:

- To the best of our knowledge, this is the first work that considers the online adaptation with varying number of heterogeneous teammates in open ad hoc teamwork;

- We propose accommodating the heterogeneity of teammates in an open AHT setting and introducing the heterogeneous graph attention network to learn teamwork situations to facilitate teamwork effectively.

- We present a formal derivation, grounded in variational inference, that makes the local teamwork situation embedding informatively consistent with global context, bolstering the theoretical foundation of our Open-OTAF designs.

- We show that the proposed method achieves superior performance on a range of challenging multi-agent tasks, including the Level Based Foraging (LBF), Wolfpack, Penalized Cooperative Navigation (PCN), and the Fortattack (Rahman et al., 2022). The extensive experiments confirm that Open-OTAF achieves dynamic teamwork for open ad hoc teams and delivers superior performance against state-of-the-art methods with a faster convergence speed.

The remainder of this paper is organized as follows. Section 2 provides a brief review of the works in MARL, ad hoc teamwork, agent modelling, and graph neural networks. Section 3 introduces the preliminaries of the proposed method. Section 4 elaborates on the detailed designs of the Open-OTAF framework. Section 5 provides the evaluation experiments and ablation studies. Finally, Section 6 concludes the paper and presents future works.

## 2    Related work

**Cooperative MARL.** Recently, the developments in MARL have led to remarkable progress in creating intelligent agents that can efficiently cooperate to solve complex tasks (Hernandez-Leal et al., 2019; Zhang et al., 2021; Gronauer & Diepold, 2022). MARL algorithms use RL techniques to train a fleet of agents in a multi-agent system. There are two main frameworks of MARL methods: centralized and decentralized learning. Centralized methods (Claus & Boutilier, 1998) learn a single policy to predict the joint actions of all agents directly, while decentralized learning (Littman, 1994) entails each agent independently. The framework of CTDE bridges the gap between the two, which permits information sharing during training, while policies are only consuming the agents' local observations, enabling decentralized execution (Lowe et al., 2017). In cooperative MARL, the policy gradient methods are a sub-class of CTDE, wherein each agent consists of a decentralized actor and a centralized critic, which is jointly optimized with the shared information of the controlled agents (Lowe et al., 2017; Foerster et al., 2018). The value decomposition methods represent the joint Q-function as a function of agents' local Q-functions (Sunehag et al., 2017; Rashid et al., 2018; Son et al., 2019; Wang et al., 2020). However, these studies presume the team configuration (e.g., team size, teammate types, team formation, and goals) stays unchanged.

**Ad hoc teamwork.** In many real-world settings, agents are required to collaborate in diverse teams without previous joint training, which comes up with ad hoc teamwork (AHT) problems. There are three main assumptions that characterise AHT (Mirsky et al., 2022). The primary assumption is the lack of prior coordination between agents, which means that the learner should be able to cooperate with the team on-the-fly, without the opportunity to rely on previously agreed collaboration strategies. The second assumption is that the learner lacks control over its teammates. Lastly, teammates are assumed to be collaborative, indicating that all agents in the team have a common goal and can take actions that will benefit the team. However, teammates might have additional objectives that may vary per teammate and even have different rewards. Early works in ad hoc teamwork operated under the assumption that the teammate's behaviour was known to the learning agent (Stone & Kraus, 2010; Agmon & Stone, 2012). Other approaches have relaxed this assumption and assumed teammate behaviour to be unknown. For instance, (Barrett et al., 2017) developed a method that infers teammates' policies from their observed behavior and subsequently incorporates these estimates into the decision-making process. Such methods typically utilise the concept of types, which encapsulates the important information determining an agent's behaviour. Existing type-based methods assume that the number of teammates is fixed and their behaviours are categorized into several known types (Barrett & Stone, 2015; Mirsky et al., 2020; Albrecht & Stone, 2019; Chen et al., 2020; Ravula, 2019). However, it is quite often that teammates with unknown types may enter and leave the environment without prior notification. The drawback of these approaches is that finite and fixed types might not cover all possible situations in complex environments (Chandrasekaran et al., 2016; Rahman et al., 2021b; 2022). Another strong assumption in these works is that the agent could access other teammates' observations and actions, which is unrealistic in partially observable environments. Though some previous arts have directed their attention toward addressing ad hoc teamwork under partial observability, they only considered closed environments in which the number of agents is fixed (Gu et al., 2021; Ribeiro et al., 2022; Schäfer et al., 2022; Zhang et al., 2023; Rahman et al., 2023). However, it is quite often that agents may join and leave the team anytime, making learning in such open AHT environments still an open research challenge. Existing

approaches neglect the heterogeneity (i.e., the varying types) among teammates and the varying team size. To bridge this gap, we propose a novel application of HAN, which simultaneously handles the potential team composition while generating semantically embeddings that capture varying team size and compositions.

**Agent modelling.** Agent modelling is a research topic that emerged alongside game theory (Brown, 1951). With the powerful representation capabilities of recent deep neural networks, the controlled agents can reason about other agents' unknown goals and behaviours (Albrecht & Stone, 2018; He et al., 2016; Albrecht & Stone, 2019). For example, the recursive reasoning method, MeLIBA, conditions the ad hoc agent's policy on a belief over teammates, which is updated following the Bayesian rule (Zintgraf et al., 2021). While these works assume that the teammates choose their actions independently. As a result, the Relational Forward Model (RFM) architecture was proposed, which can be used to model teammates' potential effects on each other (Tacchetti et al., 2019). However, the aforementioned works commonly assume access to other agents' information. Recent work proposes to use VAE to learn teammate information under partial observability teamwork (Gu et al., 2021). Building upon these developments, we propose an agent modeling framework for open ad hoc teamwork that predicts teammate behaviors by integrating joint action-value estimates with action predictions from an agent model. This approach enables optimal action selection despite uncertainty in teammate strategies.

**Graph neural network.** Graph neural networks (GNNs) are a type of powerful neural architecture that can work with graph-structured data (Wu et al., 2020), which have been exploited to learn factored action values to simplify the computation of optimal joint actions using coordination graphs (CGs) (Böhmer et al., 2020; Naderializadeh et al., 2020). Unlike these methods that use CGs to model joint action values for fully cooperative setups, some works use them in open systems to model the impact of other agents' actions towards the learning agent's returns (Rahman et al., 2021b; Jiang et al., 2019) to handle dynamic input sizes. These works have overlooked the different relationships and importance among the agents introduced by the open environment. We first introduce the heterogeneous graph to compute the ultimate teamwork situation embedding to facilitate decision-making by considering the potential effect of other agents. Together with previous works, we show the great potential of HAN, which could inspire further research.

## 3 Problem formulation

We aim to train a single autonomous agent, the ad hoc agent, that can co-operate with various teammates without pre-coordination in a partially observable open environment. The problem can be formulated by extending the framework of partially observable MDP (POMDP) (Lowe et al., 2017) to an *Open POMDP*, which is defined by a tuple $\langle \mathcal{M}, \mathcal{S}, \mathcal{O}, \mathcal{A}, \mathcal{T}, \Gamma, \mathcal{R}, \gamma \rangle$. Specifically, $\mathcal{M}$ denotes the dynamic set of agents involved in the task. $\mathcal{S}$ is a set of states describing the possible configuration of all agents and the external environment,

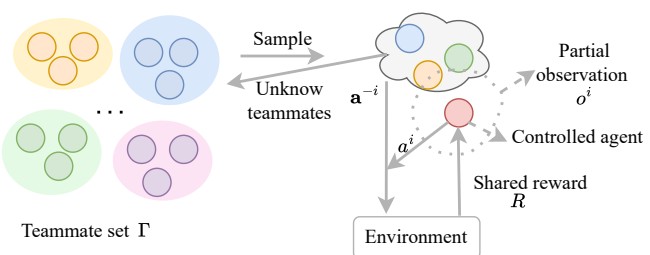

Figure 1: Visualization of the Dec-POMDP with teammate set.

where the number and type of agents vary. Each agent $k$ has its own observation space $\mathcal{O}^k \in \mathcal{O}$ with a corresponding observation function $\mathcal{O}^k : \mathcal{S} \to \mathcal{O}^k$ where $\mathcal{O}^k \in \mathcal{O}$. The agent types can be determined by their policies, which belong to a global policy set $\Gamma$.

Figure 1 shows the detailed schematics of this problem. We denote the policy of the ad hoc agent as $\pi^i$ and the joint policy of all other agents as $\boldsymbol{\pi}^{-i}$. Each agent $k$ selects an action $a^k \in A^k$ from its own action space $A^k \in \mathcal{A}$, giving rise to a joint action $[a^1, \cdots, a^n] \in A^1 \times A^2 \times \cdots \times A^n$ with $n$ being the variable number of agent. The state transits by following a transition function $\mathcal{T} : \mathcal{S} \times A^1 \times \cdots \times A^n \to \mathcal{S}$. After each transition, agent $k$ receives a new observation and obtains a scalar reward according to its reward function $r^k \in \mathcal{R} : \mathcal{S} \times A^k \to \mathbb{R}$. The initial state $s \in \mathcal{S}$ is determined by some prior distribution $p : \mathcal{S} \to [0, 1]$.

The ad hoc agent's optimal policy $\pi^{i*}$ is defined by maximizing the joint action value $Q^{\pi^i}(s, a^i, \boldsymbol{a}^{-i})$, which indicates the expected accumulative team reward over different ad hoc teamwork:

$$Q^{\pi^i}(s, a^i, \boldsymbol{a}^{-i}) = \mathbb{E}_{a^i_{t=1:+\infty} \sim \pi^i, \boldsymbol{a}^{-i}_{t=1:+\infty} \sim \boldsymbol{\pi}^{-i}, \boldsymbol{\pi}^{-i} \sim \Gamma}[\sum_{t=0}^{+\infty} \gamma_t r_t | s_0 = s, a_0 = a, P] \tag{1}$$

$$Q^{\pi^{i*}}(s, a^i, \boldsymbol{a}^{-i}) \leq Q^{\pi^i}(s, a^i, \boldsymbol{a}^{-i}), \forall \pi^i, s, a^i, \boldsymbol{a}^{-i} \tag{2}$$

Traditional MARL often optimizes team utility directly via $Q^{\pi^i}(s, a^i, \boldsymbol{a}^{-i})$, but this fails to attribute individual contributions in ad hoc teams (Gu et al., 2021). **Marginal Utility** is defined to measure the contribution of an ad hoc agent to the whole team utility (Genter et al., 2011). It represents the increase (or decrease) in a team's utility when an ad hoc agent is added to the team. Given teammates' actions $\boldsymbol{a}^{-i}$, the marginal utility can be calculated as $u^i(s, a^i, \boldsymbol{a}^{-i}) = \mathbb{E}_{\boldsymbol{a}^{-i}}[Q^{\pi^i}(s, a^i, \boldsymbol{a}^{-i}) - Q^i(\pi^i)(s, a^i_{null}, \boldsymbol{a}^{-i})]$, where $a^i_{null}$ is a baseline action (no-op or current policy output). The relationship between the marginal utility and the team utility (denoted by the joint action value) can be calculated as: $\arg\max_{a^i} u^i(s, a^i, \boldsymbol{a}^{-i}) = \arg\max_{a^i} Q^{\pi^i}(s, a^i, \boldsymbol{a}^{-i})$, where $u^i(s, a^i, \boldsymbol{a}^{-i})$ denotes the marginal utility when the ad hoc agent chooses the action $a^i$ under the state $s$. The agent chooses the action that maximizes the marginal utility to ensure the maximal team utility.

To enable adaptive decision-making, the ad hoc agent's policy must respond to teammate behaviors. We address the complexity of unknown teammates by encoding their interactions into a teamwork situation—a compact latent representation of the team's strategic state. This representation captures how teammates collectively influence environmental dynamics, allowing the agent to optimize its behavior for the current cooperative context. For example, in our truck-drone delivery system, the teamwork situation embedding encodes the state of task phases, interaction patterns and so on. To model the concept clearly, we formally define the teamwork situation as:

**Definition 1.** *(Teamwork situation) At each time step $t$, the ad hoc agent is in the teamwork situation $C^i_t \in C$, which is the current underlying teamwork state yielded by the environment state $s_t$ and other teammates' actions $\mathbf{a}^{-i}_t$. It reflects the high-level semantics about the teammates' behaviours.*

Though different teammates generate diverse state-action trajectories, they may lead to similar teamwork situations at certain moments, and the ad hoc agent's action would affect their transitions (Gu et al., 2021). By accurately identifying the current teamwork situation, the ad hoc agent can choose the action accordingly to ensure online adaptation. In this work, the dynamic teammate mainly refers to the different combinations and numbers of teammates. At the beginning of each episode, the teammates' types are randomly sampled and kept fixed in that episode. However, the teammate combination and number are different across episodes. In addition, the set of behaviour types used for sampling is different for training and evaluation.

## 4 Methodology

### 4.1 The overall framework

Open-OTAF aims to estimate the ad hoc agent's marginal utility for choosing corresponding actions to maximize the team utility. To enable adaptation to unknown teammates, we model the marginal utility as a conditional function on the inferred latent variable, which implicitly represents the current teamwork situation. Open-OTAF jointly optimizes the marginal utility function and the latent variable by two learning objectives in an end-to-end fashion. Figure 2 depicts the overall framework of our Open-OTAF algorithm. It splits the team into two parts: teammates and the ad hoc agent.

We regard other teammates as a part of the environmental dynamics perceived by the ad hoc agent. Different combinations of teammates lead to diverse and complex dynamics. We expect to learn teamwork situations to describe the core information of teammates' behaviours implicitly. As similar teammates might possess similarities in their identifications, learning a specific context for each teammate but ignoring the relationships among them could lead to trivial encodings. We thus assign them to different clusters via the Chinese Restaurant Process (CRP) to shrink the context search space (Section 4.2 and Figure 3(a)). Then, we

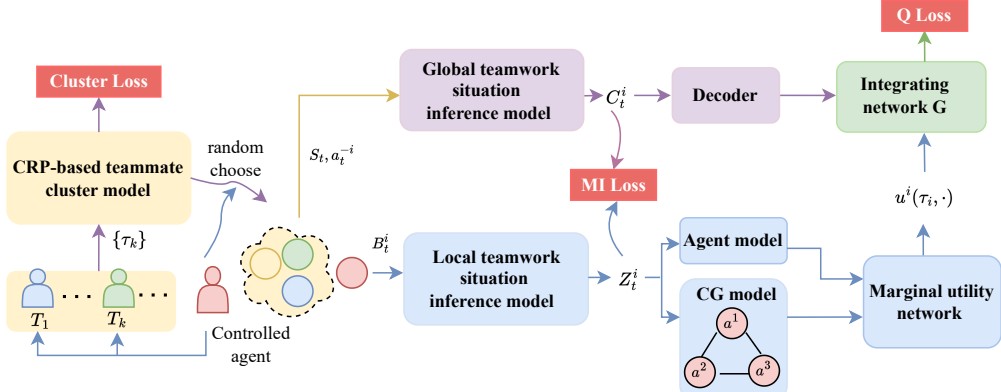

Figure 2: Overall framework of Open-OTAF. In training phase, we introduce a global teamwork situation inference model to obtain $C_t^i$. Then, an integrating network $G$ and a teamwork situation decoder are jointly proposed to regularize the information embedded in the learned variable. For controlled agent, we introduce a local teamwork situation inference model to infer a proxy representation $Z_t^i$ of $C_t^i$ from local observation.

introduce a global teamwork situation inference model that takes the controlled agent's state, the actions of its teammates, and the estimation of the team's composition as input. This inference model processes these inputs to infer and output the current teamwork situation $C_t^i$, enabling a more informed decision-making process. However, the size of the obtained embedding is changeable due to the variable team size. To handle the dynamic team size and explicitly model the complex relationships among agents, we introduce HAN as an effective function approximation model to obtain the fixed-length embedding $C_t^i$, which can be optimized by the loss function (Section 4.3 and Figure 3(c)).

For the controlled agent, we expect to condition its policy on the current teamwork situation $C_t^i$. As the partial observability impedes the direct access to $C_t^i$, we introduce a local teamwork situation inference model to infer a proxy representation $Z_t^i$ of $C_t^i$ from local observation. We force $Z_t^i$ to be informationally consistent with $C_t^i$ by an information-based loss function (MI loss) to maximize the conditional mutual information. To select optimal actions from the joint action value model, we propose action-value computation method which implements the joint action value model as fully connected Coordination Graphs (CG) with action predictions learned using an agent model. Then, we train a marginal utility network to estimate the ad hoc agent's conditional marginal utility (Section 4.4). Similar to the CTDE scenario, we relax the partial observability in the training phase. Open-OTAF is granted access to the global state $s_t$ and other teammates' actions during training. For the controlled coordinating policy training, we sample representative teammates to coordinate with by capturing their identifications into distinguishing contexts to augment the joint policy during the centralized training phase. Each agent then utilizes its local information to approximate the global context information. The mentioned processes proceed alternately, and we can finally obtain a robust policy to adapt to any teammates gradually during the decentralized execution phase. We show the meaning of the notations in Table A.1.

## 4.2 CRP-based teammate cluster model

In this section, we introduce the CRP-based teammate cluster model in detail. When identifying the teamwork situation, it is irrational and inefficient to treat each new teammate as a novel type ignoring their similarities. Accordingly, we aim to acquire clearly distinguishable boundaries of teammates' behaviours by applying a behaviour detection module to assign teammates with similar behaviours to the same cluster.

As shown in Figure 3(a), we have a batch of training teammates with the corresponding trajectories $\{\{\tau^1\}, \{\tau^2\}, \cdots, \{\tau^k\}, \cdots\}$ sampled from the interaction between the $k^{th}$ teammate and the environment. The trajectory set $\{\tau^k\}$ is a set of trajectories $\tau = (s_0, a_0, \cdots, s_T)$ sampled from the interactions between the $k^{th}$ teammate and the environment, and $T$ denotes the time horizon. Given the high dimensionality

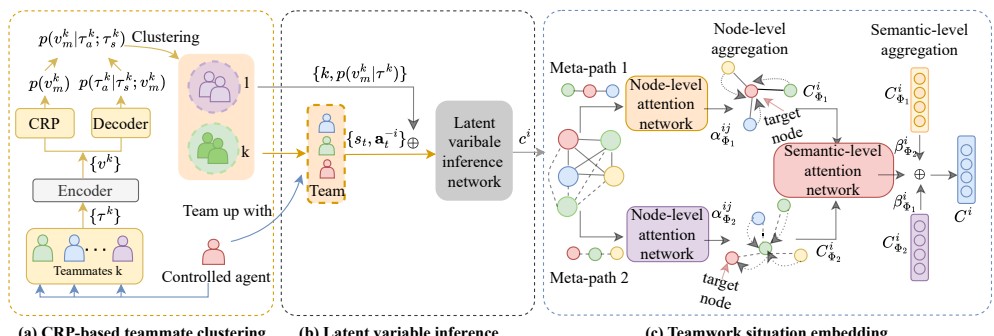

**(a) CRP-based teammate clustering**  **(b) Latent variable inference**  **(c) Teamwork situation embedding**

Figure 3: Global teamwork situation inference model: we first design a teammate cluster model to acquire clearly distinguishable boundaries of teammates' behaviours and formulate the distribution of dynamic teammates. Then, we introduce a variational auto-encoder to get the teamwork situation by taking the global state $S$ and teammate actions $\boldsymbol{a}^{-i}$ as input. The current teamwork situation can be encoded by variable-size latent variables $c^i$. Then, we introduce the HAN to learn a fixed-size embedding of the teamwork situation.

of the trajectories, we use a trajectory encoder $E_{\omega_1}$ to extract features from the trajectory $\tau$ and represent it as $v = E_{\omega_1}(\tau)$, where $\omega_1$ is the parameter of the trajectory encoder $E$. Concretely, we partition the trajectory $\tau$ into $\tau_s = (s_0, s_1, \cdots, s_{T-1}, s_T)$ and $\tau_a = (a_0, \cdots, a_{T-1})$. For the $k^{th}$ teammates generate so far, $v^k = \mathbb{E}_{\tau^k \sim \{\tau^k\}}[E_{\omega_1}(\tau^k)]$ will be used to represent its behavioural type.

If $M$ clusters are instantiated so far, the cluster that the $k^{th}$ teammate belongs to will be inferred by $P(v_m^k|\tau^k) = P(v_m^k|\tau_s^k, \tau_a^k), m \in \{1, \cdots, M, M+1\}$. The $v_m^k$ is a variable indicating whether the $k^{th}$ teammate belongs to the $m^{th}$ cluster based on its representation $v^k$. Based on the Bayesian rules (Blei & Frazier, 2011), the posterior distribution $P(v_m^k|\tau_s^k, \tau_a^k)$ can be decomposed as $P(v_m^k|\tau_s^k, \tau_a^k) \propto P(v_m^k)P(\tau_a^k|v_m^k; \tau_s^k)$. The prior $P(v_m^k)$ is given by CRP, representing the probability of the $k^{th}$ teammate belonging to $m^{th}$ cluster without the teammate strategy information. If $M$ clusters are formed so far, the prior $P(v_m^k) = n_m/(k-1+\lambda)$ when $m \leq M$, else $P(v_m^k) = \lambda/(k-1+\lambda)$ when $m = M+1$, where $n_m$ denotes the number of teammate belonging to the $m^{th}$ cluster, $\sum_{m=1}^{M} n_m = k-1$ and $\lambda > 0$ is a concentration hyperparameter that controls the probability of the formation of a new cluster.

To estimate the conditional probability $P(\tau_a^k|\tau_s^k, v_m^k)$, we use a parametric network $D_{\omega_2}$ that take $\tau_s^k, v_m^k$ as input to estimate the likelihood of taking action $\tau_a^k$, i.e., $P(\tau_a^k|v_m^k; \tau_s^k) = D_{\omega_2}(\tau_a^k|\tau_s^k; v_m^k)$. Then $v_m^k$ can be update by $v_m^k = \frac{n_m \bar{v}_m + v^k}{n_m + 1}$ when $m \leq M$, else $v_m^k = v^k$ if $m = M+1$, where $\bar{v}_m$ is the mean value of the $m^{th}$ cluster. Combining the estimated prior distribution and predictive likelihood, we are able to decide which cluster the $k^{th}$ teammate belongs to and their corresponding probability, that is $m* = \arg\max_m P(v_k^m|\tau_k)$. If $m* \leq M$, we will assign the $k^{th}$ teammate to the $m^{th}$ cluster. Then, cluser center $\bar{v}_{m*}$ will be updated by $\bar{v}_{m*} = \frac{n_{m*} \bar{v}_{m*} + v^k}{n_{m*} + 1}$ when $m* \leq M$, else $\bar{v}_{M+1} = v^k$ if $m = M+1$. To force the learned representation $v$ to capture the behavioural information of each teammate and estimate the predictive likelihood more precisely, the $E_{\omega_1}$ and $D_{\omega_2}$ can be optimized by:

$$\mathcal{L}_C(\boldsymbol{\omega}) = \mathbb{E}_{\tau \sim \bigcup_{k=1}^{K} \mathcal{D}^k} - \log[D_{\omega_2}(\tau_a|\tau_s; E_{w_1}(\tau)), \tag{3}$$

where $K$ is the number of teammates and $\boldsymbol{\omega}$ is the weight of the encoder and decoder networks.

### 4.3 Learning to represent teamwork situation

After clustering teammates into behaviorally distinct groups, we train a robust policy capable of adapting to dynamic team compositions by conditioning the controllable agent's actions on inferred teammate behaviors. Despite the diversity and complexity that unknown teammates' behavior exhibits, the CRP formalized before helps acquire clearly distinguishable boundaries based on teammates' behavioral types with regard to high-level semantics. Furthermore, motivated by the ODTIS (Gu et al., 2021), which aims to guide the context

encoder to identify the adaptive behaviors, we expect to utilize a global context encoder to embed the teammates' information into a compact but descriptive representation.

As shown in figure 3(b), we encode the teamwork situation in a stochastic embedding space via a latent variable inference network, which takes trajectory $\tau_t^m = (s_0^m, a_0^m, \cdots, s_t^m)$ and their cluster information $\{k, p(v_m^k|\tau^k)\}$ as input, which emerges from the interactions between paired joint policy $(\pi, \bar{\pi}^m)$ and the environment, and $\bar{\pi}^m$ is the joint policy of uncontrollable teammates belonging to the $m^{th}$ cluster. For clarity, we omit time step $t$ in all formulas below. To solve this issue, the current teamwork situation can be encoded by latent probabilistic variables $c^i$ drawn from a multivariate Gaussian distribution $\mathcal{N}(\mu_{c^i}; \sigma_{c^i})$. To enable the dependency revealed in the definition, we use a trainable neural network $f$ to learn the parameters of the Gaussian distribution of $c^i$ by $(\mu_{c^i}, \sigma_{c^i}) = f(\tau^m, \{k, p(v_m^k|\tau^k)\}; \theta_f), c^i \sim \mathcal{N}(\mu_{c^i}, \sigma_{c^i})$, where $\theta_f$ are the parameters of $f$. Note that the size of $c^i$ corresponds to the number of teammates, which can change in the open ad hoc teamwork. Thus, we introduce the heterogeneous attention graph neural network, using the inherent ability of the graph neural network to transform the dynamic-size teammate's information into a fixed-size output. In heterogeneous graphs, meta-paths are predefined composite relations that encode high-order semantic relationships between nodes through sequences of edge types. The HAN leverages these meta-paths in two aspects: (1) it computes relationship-specific attention weights for each meta-path, enabling the model to dynamically prioritize relevant interactions, and (2) it aggregates node embeddings along each meta-path, producing a final representation as a weighted combination of these semantically distinct pathways. This dual-process architecture enables HAN to model complex relational patterns while preserving interpretability through meta-path-specific attention distributions.

We denote the heterogeneous graph as $H = (X, \mathcal{E})$, where $X = \{X_i | \forall i \in \{1, \cdots, n\}\}$ is a set of nodes representing the team agents, and $\mathcal{E}$ is the set of symmetric edges between every node pair. We use a HAN with various types of edges to model open ad hoc teamwork. The reason is that the agent number is variable and the relationships among agents are different in open ad hoc teamwork. As shown in Figure 4, the controlled agents may not construct a cooperation relationship with the teammates planning to leave or just entering the environment, and the influence of teammates on the controlled agent may be different. We model the interaction relationship between the controlled agent and different teammates via different meta-paths, i.e., $\Phi = \{cooperate, no\ cooperate\}$.

As shown in Figure 3(c), each meta-path has its own attention network and learns the attention weights $\alpha_{\Phi_p}^{ij}$ for the neighboring agents of the corresponding relationship. For instance, the "cooperate" meta-path only takes the node embedding of the neighboring agents that cooperate with the controlled agent as input. Thus, different meta-paths have diverse semantic information. The network is implemented as a two-layer MLP with: $e_{\Phi_p}^{ij} = MLP(W_{\Phi_p}[c^i \| c^j])$, where $W_{\Phi_p}$ is learnable weight matrix for meta-path $\Phi_p$, $c^i$ and $c^j$ are node embedding and $\|$ denotes concatenation. After that, we normalize them

Figure 4: Heterogeneous graph representation of open ad hoc teamwork.

to get the weight coefficient $\alpha_{\Phi_p}^{ij} = softmax_j(e_{\Phi_p}^{ij})$. Then, the learned embedding of node $i$ for the meta-path $\Phi_p$ can be calculated as $C_{\Phi_p}^i = g(\sum_{j \in \mathcal{N}_{\Phi_p}^i} \alpha_{ij}^{\Phi_p} \cdot c^j)$, where $\mathcal{N}_{\Phi_p}^i (p \in \{1, 2\})$ denotes the meta-path $p$ based neighbors of node $i$, and $g$ denote the activation function. Then, the final teamwork situation embedding is aggregated by all semantic embeddings, where the semantic level attention network is used to learn the weight $\beta_{\Phi_p}^i$ of different meta-paths can be calculated by $\beta_{\Phi_p}^i = exp(MLP(C_{\Phi_p}^i)) / \sum_{p=1}^{2} exp(MLP(C_{\Phi_p}^i))$. We can fuse these semantic-specific embeddings to obtain the final embedding as $C^i = \sum_{p=1}^{2} \beta_{\Phi_p}^i \cdot C_{\Phi_p}^i$. Our model can get the optimal combination of neighbors and multiple meta-paths in a hierarchical manner, which enables $C^i$ to better capture the complex structure and rich semantic information.

We then use a marginal utility network to estimate the ad hoc agent's marginal utility as $\hat{u}^i(\tau^i, a^i; C^i) \approx u^i(s, a^i, \boldsymbol{a}^{-i})$, representing its incremental contribution to the team's expected return, conditioned on the current teamwork context. This quantifies how much $a^i$ improves team performance, given teammates' behaviors encoded in $C^i$. After that, a loss function (Q loss), an integrating network $G$ and a teamwork situation decoder $g$ (as shown in Figure 2) are jointly proposed to regularize the information embedded in the learned variable $C^i$. The integrating network $G$ generates the joint action value's estimation as $G(u^i, C^i) \approx Q^{\pi^i}(s, a^i, \boldsymbol{a}^{-i})$, where $G$ maps the ad hoc agent's utility $u^i$ into the value estimation. Given any $C^i$, the increase of the ad hoc agent's marginal utility results in the improved joint action value. We adopt a modified asynchronous Q-learning loss function (Q-loss) (Mnih et al., 2016) as the optimization objective:

$$\mathcal{L}_Q = \mathbb{E}_{u_t^i, C_t^i, r_t \sim \mathcal{D}}[[r_t + \gamma \max_{a_{t+1}^i} \bar{G}(u_{t+1}^i, C_{t+1}^i) - G(u_t^i, C_t^i)]^2] \tag{4}$$

where $\bar{G}$ is a periodically updated target network. The expectation is estimated with uniform samples from the replay buffer $\mathcal{D}$.

## 4.4 Learning marginal utility function under partial observability

For ad hoc agents, we aim to model the marginal utility as a conditional function on the inferred teamwork situations. However, partial observability impedes the agent's access to $C^i$ encoded from the global state-action trajectory. Thus, we introduce a local inference model $f^*$ to derive proxy representations of the teamwork situation $c^i$ from local observations $B_t^i = (o_t^i, r_{t-1}, a_{t-1}^i, o_{t-1}^i)$. We assume that $B_t^i$ can partly reflect the current teamwork situation since the transition implicitly indicates the underlying dynamics, which is primarily influenced by other teammates' behaviors. We denote the estimation of $c^i$ as $z^i$. Similar to $c^i$, we encode $z^i$ into a stochastic embedding space as $(\mu_{z^i}, \sigma^i) = f^*(B^i; \theta_{f*}), z^i \sim \mathcal{N}(\mu_{z^i}, \sigma_{z^i}),$, where $\theta_{f*}$ are parameters of $f^*$. Then, with the predicted teammate type information, we obtain the final teamwork situation embedding $Z^i$ via the HAN. To make $Z^i$ informatively consistent with $C^i$, we introduce an information-based loss function $\mathcal{L}_{MI}$ to maximize the conditional mutual information $I(Z^i; C^i|B^i)$ between $Z^i$ and $C^i$. Due to the difficulty and feasibility of estimating the mutual information, a variational distribution $q_\xi(Z^i|C^i, B^i)$ is used to approximate the conditional distribution $p_\xi(Z^i|C^i, B^i)$. Inspired by the information bottleneck (Alemi et al., 2016), we derive a tractable lower bound for the $MI$ as follows.

**Theorem 4.1.** *Let $I(Z^i; C^i|B^i)$ be the conditional mutual information between the local teamwork situation embedding $Z^i$ of agent i and global embedding $C^i$. Then, the lower bound is given by:*

$$I(Z^i; C^i|B^i) \geq \mathbb{E}_{Z^i, C^i, B^i}\left[\log \frac{q_\xi(Z^i|C^i, B^i)}{p(Z^i|B^i)}\right]. \tag{5}$$

The lower bound can be rewritten as a loss function: $\mathcal{L}_{MI} = \mathbb{E}_{Z^i, C^i, \tau^i \sim \mathcal{D}}[D_{KL}[p(Z^i|B^i)||q_\xi(Z^i|C^i, B^i)]]$, where $\mathcal{D}$ is the replay buffer.

Notice that the entropy of our labels is independent of our optimization procedure and so can be ignored. The proxy encoder is conditioned on the transition data. Given the transitions, the distribution of the proxy representations $p(Z^i)$ is independent of the local histories. Thus, we have:

$$I(Z^i; C^i|B^i) \geq -\mathbb{E}_{C^i, B^i}[\mathcal{CE}[p(Z^i|C^i, B^i)||q_\xi(Z^i|C^i, B^i)]dZ^i].$$

In practice, we sample data from the replay buffer, and the lower bound can be rewritten as a loss function to be minimized by:

$$\mathcal{L}_{MI} = \mathbb{E}_{Z^i, C^i, B^i}[D_{KL}[p(Z^i|B^i)||q_\xi(Z^i|C^i, B^i)]]. \tag{6}$$

To achieve a robust policy, we use the inferred $Z^i$ to estimate the agent's marginal utility $\mu(Z^i, a^i)$. Considering the influence of other teammates on the controlled agent, we adopt the coordination graphs (CGs) (Rahman et al., 2021b) to estimate the ad hoc agent's marginal utility. Given the embedding $Z^i$, the agent

modelling module computes the likelihood of the agent's actions $p(a^j|Z^i)$. Then, the marginal utility of agent $i$ can be calculated as:

$$
\begin{aligned}
\mu(Z^i, a^i) = Q^i(a^i|Z^i) + \sum_{j \in N, j \neq i}(Q^j(a^j|Z^i) + Q^{i,j}(a^i, a^j|Z^i))p(a^j|Z^i) \\
+ \sum_{j,k \in N; j,k \neq i} Q^{j,k}(a^j, a^k|Z^i)p(a^j|Z^i)p(a^k|Z^i).
\end{aligned}
\tag{7}
$$

$Q^j(a^j|Z^i)$ represents $j's$ contribution towards the learner's returns by executing $a^j$, while $Q^{j,k}(a^j, a^k|Z^i)$ denotes $j$ and $k's$ contribution towards the learner's returns by jointly choosing $a^j$ and $a^k$. We implement $Q^j(a^j|Z^i)$ and $Q^{j,k}(a^j, a^k|Z^i)$ as multilayer perceptrons (MLPs) parameterized by $\beta$ and $\delta$ to enable generalization across different numbers of the agents. Afterward, the marginal utility can be used to approximate the joint action-value function. Therefore, the overall optimization objective can thus be derived as $\mathcal{L}_{tot} = \mathcal{L}_Q + \mathcal{L}_{MI}$. During the training phase, the controlled agent interacts with different teammates to collect transition data into the replay buffer. Then, samples from $\mathcal{D}$ are fed into the framework for updating all parameters induced by the overall loss. During execution, the controlled agent conditions its behaviour on the inferred teamwork situations by choosing actions to maximize the utility function. We summarize our training procedure in Algorithm 1 in the Appendix A.1.

## 5 Experiment

To thoroughly assess the Open-OTAF algorithm, we evaluate our model in four widely-used partial observable multi-agent environments for various settings, including the Level-Based Foraging (LBF), Wolfpack, the Penalized Cooperative Navigation (PCN), and the Fortattack (Rahman et al., 2022). From the experiments, we aim to answer the following questions: 1) Can Open-OTAF yield superior performance than SOTA baseline methods (Figure 5)? 2): Are the main components, i.e., the CRP, MI, and AM, necessary and effective (Figure 6, Figure A.5)? 3) Does Open-OTAF have generalization capacity with the varying teammate size and type (Figure A.6 and Table 1)? 4): How do the key hyperparameters, cluster numbers, affect the final results (Table A.4)?

### 5.1 Experiment Setup and baselines

We select four multi-agent tasks as our environments, as shown in Figure A.2 in the Appendix A.2.1. Among them, Level Based Foraging (LBF), Wolfpack and Fortattack are three scenarios coming from (Rahman et al., 2021b). LBF is a cooperative grid world game with agents that are rewarded if they concurrently navigate to the food and collect it. In Wolfpack, multiple agents (predators) need to chase and encounter the adversary agent (prey) to win the game. The FortAttack environment defines a bounded two-dimensional space where agents are constrained within specific coordinate ranges. We also evaluate our method in the penalized cooperative navigation, coming from the MPE environment (Lowe et al., 2017), where multiple agents are trained to move towards landmarks while avoiding collisions with

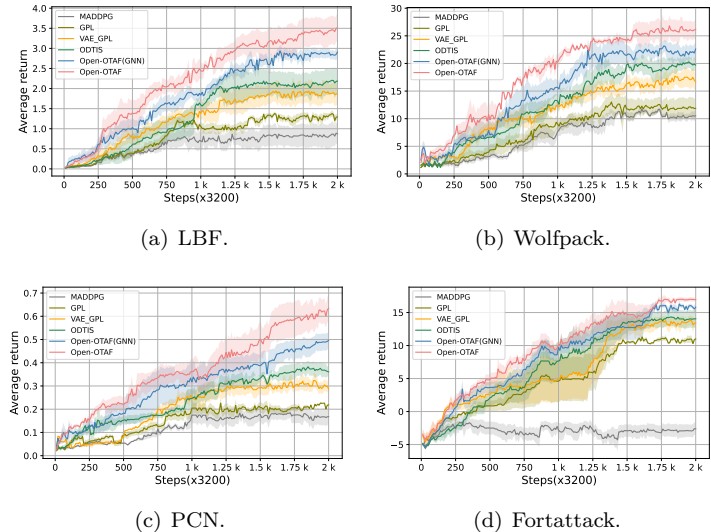

(a) LBF.  (b) Wolfpack.

(c) PCN.  (d) Fortattack.

Figure 5: Performance comparison across various scenarios. For LBF, Wolfpack, PCN and Fortattack, the training steps are 640, 0000 and we evaluate all the methods with 100 test episodes after every 3200 training steps.

each other. We simulate open environments by creating an open process that determines how agents enter and leave during episodes, for both training and testing. In the LBF, Wolfpack, and PCN environments, the number of timesteps an agent can exist in the environment is determined by uniformly sampling from a certain range of integers. In FortAttack, agents are removed once they are shot by opponents. After being removed, agents can reenter the environment after a specific period of waiting time. The type of an agent entering an environment is uniformly sampled from all available types. More details about the environments and algorithm configuration are provided in the Appendix A.2.1 and A.3.

To illustrate the advancement of our method, we compare the proposed method with two type-based baselines, GPL and VAE-GPL, which can be used to solve open ad hoc teamwork under full and partial observability separately. Although GPL assumes full observability of the state, we apply GPL in our experiments by treating the learner's observations as input. ODITS (Gu et al., 2021) applies a centralized "teamwork situation encoder" for end-to-end learning to adapt to arbitrary teammates in closed environments. As ODITS considers only a single ad hoc agent in the closed environment, we further extend the ODTIS to open environments by introducing a graph neural network. Lastly, to demonstrate the advantages of HAN, we also compare the algorithm Open-OTAF(GNN), where GNNs are introduced to produce the representations of teamwork situations. Table A.2 (in Appendix A.2.4) provides a summary of the different algorithms.

## 5.2 Experiment results

### 5.2.1 Analysis of the average return

We first compare the overall performance of Open-OTAF with baselines. We report the performance in terms of average returns (solid line) and the standard deviation (shaded areas) with 7 random seeds, as summarized in Figure 5 and Table 1. From the curves, we observe that the returns of Open-OTAF consistently outperform the other four baselines with a faster convergence speed in all benchmark games while possessing a lower variance, which basically shows the effectiveness and high efficiency of the proposed method. In particular, the comparison with Open-OTAF(GNN) suggests the effectiveness of using HAN to produce representations of the teamwork situation. The utilization of HAN can boost the performance of open ad hoc teamwork than conventional GNN. Otherwise, the comparison between Open-OTAF and ODTIS shows that using the coordination graphs and agent model for optimal marginal utility estimation can boost performance as well. It enables our method to estimate the marginal utility by considering the effect of other agents, which leads to significantly better training. Also, ODITS shows superior performance than GPL and VAE-GPL, manifesting the necessity of utilizing global states to improve training efficiency. Furthermore, GPL performs poorly than other methods because the sequence of observations perceived by the learner contains the least useful information compared to other methods. These results show that our method can design an autonomous agent capable of robust teamwork under dynamic teammates without pre-coordination.

### 5.2.2 Ablation studies

To determine the importance of each component of Open-OTAF, we conduct an ablation study for the different components to see how it affects the training, and the results over 5 random seeds are shown in Figure 6. We take the LBF as an example. More ablation studies are provided in the Appendix A.4. First, to illustrate the importance of the CRP-based teammate cluster, we derive the method of $w/o$ [1] CRP by removing the CRP process and taking each new teammate as a new cluster. Furthermore, we pick up $w/o$ MI to investigate how maximizing mutual information between global and local information accelerates learning efficiency. Finally, $w/o$ AM is introduced to illustrate the impact of other team-

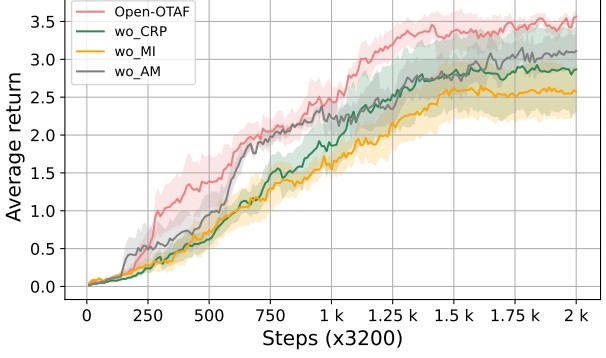

Figure 6: Ablations for different components.

---

[1] We use the $w/o$ in place of without for conciseness.

mates in the calculation of the marginal utility. We only utilize a network to calculate the marginal utility without considering the effect of other agents.

In Figure 6, the comparison results show that $w/o$ MI suffers the most severe performance degradation in all games, which demonstrates the necessity of utilizing global information to facilitate the learning of the local teamwork situation inference model. Furthermore, $w/o$ CRP achieves lower performance than Open-ATAF, which manifests that the CRP-based teammate model helps the controlled agent adapts to dynamic teammates rapidly. We also find agent modelling helps learn more accurate marginal utility by considering the effect of other teammates, which can bring about a slight coordination improvement. It can also demonstrate that the cluster model can help to learn a robust ad hoc agent. The testing results are also provided in Figure A.5, where we provide the average return (bars) and the standard deviation (error bars) over different seeds. Experimental results demonstrate that our CRP-based teammate classification model, when augmented with global information during training, enables agents to infer significantly more accurate teamwork situation representations compared to baseline methods. This improvement stems from two key mechanisms, the first one is that the CRP's ability to probabilistically cluster teammates by behavioral similarity, which reduces uncertainty in type estimation, and the integration of global state information during centralized training, which provides necessary context for disambiguating observationally similar but strategically distinct teammate behaviors.

### 5.2.3 The generalization capacity

To assess the generalization ability of Open-OTAF, we performed experiments involving different teammate numbers and types. First, we evaluate the generalization ability by changing the teammate types during testing. Similar to the training procedure of Open-OTAF, we control one agent and test it with teammates randomly sampled from a pool consisting of 10 different policies, which were not seen during training. The generalization performance of Open-OTAF and the baselines is provided in Table 1. We show the average and 95% confidence bounds during testing utilizing 5 seeds. The data was gathered by averaging the returns at the checkpoint which achieved the highest average performance during training. We highlight in bold the algorithm with the highest average returns. The results show that our method significantly outperforms other baselines during testing. Unsurprisingly, methods that achieve high returns during training, will also achieve better performance when generalizing to different teammate types. Open-OTAF's performance in this diverse teammate-type setting was significant, outclassing the baseline performances by $5.1\% \sim 49.8\%$.

Table 1: Results of generalization capacity under different teammate types.

| Env | MADDPG | GPL | VAE_GPL | ODTIS | Open-OTAF(G) | Open-OTAF |
|---|---|---|---|---|---|---|
| LBF | $0.73_{(0.62)}$ | $1.85_{(0.51)}$ | $2.14_{(0.47)}$ | $2.17_{(0.45))}$ | $2.39_{(0.31)}$ | $\mathbf{2.67_{(0.25)}}$ |
| Wolf | $8.07_{(1.25))}$ | $16.9_{(1.24))}$ | $19.3_{(1.39)}$ | $21.4_{(1.48)}$ | $24.3_{(1.42)}$ | $\mathbf{26.9_{(1.26)}}$ |
| | $0.11_{(0.15)}$ | $0.43_{(0.12)}$ | $0.52_{(0.11)}$ | $0.69_{(0.11)}$ | $0.77_{(0.10)}$ | $\mathbf{0.86_{(0.09)}}$ |
| Fortattack | $0.12_{(0.26)}$ | $0.44_{(0.22)}$ | $0.49_{(0.22)}$ | $0.54_{(0.28)}$ | $0.58_{(0.22)}$ | $\mathbf{0.62_{(0.11)}}$ |

Moreover, to assess Open-OTAF's generalization ability further, we conduct an expanded experiment involving different teammate sizes. Open-OTAF was initially trained on 5 teammates with 10 different training policies and then tested on 3, 5, 6, 8 and 10 teammates with 10 different testing teammate policies. Results, summarised in Figure A.6 (in Appendix A.4), indicate a consistently superior performance of Open-OTAF. Experimental results show that, with desirable generalization to various teammate sizes, our method outperforms the state-of-the-art method. In summary, when encountering unknown teammates, the CRP-based module first classifies them into one of the 10 clusters (we set the maximum of cluster number as 10). Then, the attention network uses the clustering results as the semantic information to generate weights for better teamwork situation estimation. The teamwork situation is the input of the controlled agent's policy and affects its action selection. We admit that we may not train a best-response policy for unknown teammates. It is also impossible to train a best-response policy for every possible unknown teammate. However, the trained clustering ability of the CRP-based module and the generalization capability of the attention network allows the controlled agent to select reasonably good actions. The ablation studies, as shown in Figures A.6,

and Table 1 demonstrate promising outcomes. It can be noted that the CRP-based teammate cluster module and the heterogeneous attention graph neural network (HAN) can improve generalization performance.

## 6 Conclusion

This work proposes a novel online adaptation reinforcement learning algorithm to address the challenging open ad hoc teamwork problem. In Open-OTAF, the CRP-based teammate cluster model aims to predict the probability distributions of the teammate's types, which can be considered as the agent's beliefs about the current status of its team. The inferred teammate types with environment state are fed into a representation learning model to encode the teamwork situation representations as latent probabilistic variables. With the assistance of the heterogeneous attention graph, we manage to infer the teamwork situation and adapt to the new teammate effectively. We also propose an information-based proxy encoder to infer the learned variables from local observations to overcome partial observability. With the guide of the global teamwork situation, the ad hoc agent adapts to new teammates' behaviours dynamically and quickly by conditioning its policy on the inferred variables. Extensive experiments demonstrate that Open-OTAF not only obtains a higher average return but also achieves a faster convergence speed. Our method can be seen as a primary attempt for online teammate adaptation with dynamic teammates in open ad-hoc teamwork, we sincerely hope it can be a solid foothold for applying RL to practical applications. For future work, we will relax the CTDE assumption to enhance the applicability of our approach, investigating a robust ad hoc agent within a fully decentralized framework, and validating the method in real-world applications.

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

Table A.1: Notations of algorithm

| Notation | Description | Notation | Description |
|---|---|---|---|
| $\lambda$ | Concentration parameter in CRP cluster model. | $G$ | Integrating network. |
| $\mathcal{O}$ | Observation space. | $\mathcal{T}$ | Transition function. |
| $\mathcal{S}$ | State space. | $E_{\omega_1}$ | Trajectory encoder. |
| $\mathcal{A}$ | Action space. | $D_{\omega_2}$ | Trajectory decoder. |
| $\pi^i$ | Policy of the controlled agent. | $\bar{v}_m$ | Mean value of the $m^{th}$ cluster. |
| $\boldsymbol{\pi}^{-i}$ | Joint policy of all other agents. | $X$ | Set of nodes. |
| $a^i$ | Action of controlled agent. | $\mathcal{E}$ | Set of edges. |
| $\boldsymbol{a}^{-i}$ | Joint action of all other agents. | $\Gamma$ | The global policy set. |
| $o^i$ | Local observation of agent $i$. | $\alpha_{ij}$ | Weight for node pair $i, j$. |
| $u^i$ | Marginal utility of agent $i$. | $\beta^i_{\Phi^p}$ | Weight for meta-path. |
| $B^i_t$ | Batch of local transition data. | $\Phi$ | Set of Meta-path. |
| $M$ | Number of clusters | $T$ | Time horizon |
| $\tau^k$ | Trajectory which sampled from the interaction between the $k^{th}$ teammate and environment. | | |
| $v^k$ | The learned embedding represents the behavioural type of $\tau^k$. | | |
| $v^k_m$ | Variable of the $k^{th}$ teammate belongs to the $m^{th}$ cluster based on its representation $v^k$. | | |
| $n_m$ | The number of teammates belonging to the $m^{th}$ cluster. | | |
| $C^i_t$ | Teamwork situation inferred by global information. | | |
| $e^{ij}_{\Phi_p}$ | Importance of meta-path based node pair $i, j$. | | |
| $Z^i_t$ | Teamwork situation inferred by local information. | | |
| $\mathcal{M}$ | Dynamic set of agents involved in the task. | | |

## A Appendix

### A.1 Details about Derivation

**Details about CRP and derivation of cluster assignment** Chinese restaurant process (CRP) (Blei & Frazier, 2011) is a discrete-time stochastic process that defines a prior distribution over the cluster structures. It can be described simply as follows: as each customer enters a restaurant with an infinite number of tables, he chooses to sit down alone at a new table with a probability proportional to a concentration parameter or sits with other customers with a probability proportional to the number of customers sitting on the occupied table. Customers sitting at the same table will be assigned to the same cluster. Motivated by the Gopalan et al. (2014), the cluster assignment of the $k^{th}$ teammate $P(v^k_m|\tau^k_s, \tau^k_a)$ can be decomposed as:

$$
\begin{aligned}
&P(v^k_m|\tau^k_s, \tau^k_a) \\
&= \frac{P(v^k_m, \tau^k_s, \tau^k_a)}{P(\tau^k_s, \tau^k_a)} \\
&= \frac{P(\tau^k_s, \tau^k_a|v^k_m)P(v^k_m)}{P(\tau^k_s, \tau^k_a)} \\
&= \frac{P(\tau^k_a|v^k_m, \tau^k_s)P(\tau^k_s|v^k_m)P(v^k_m)}{P(\tau^k_s, \tau^k_a)} \\
&\propto P(\tau^k_a|v^k_m, \tau^k_s)P(\tau^k_s|v^k_m)P(v^k_m).
\end{aligned}
\tag{8}
$$

As $\tau^k_s$ is a set of states that is not determined by the behavioural type of the teammates if neglecting the correlation in time dimensionality, $P(\tau^k_s|v^k_m)$ is a constant. Accordingly, we can obtain that $P(v^k_m|\tau^k_s, \tau^k_a) \propto P(v^k_m)P(\tau^k_a|v^k_m; \tau^k_s)$.

**Variational Bound of teammates context approximation** To make context vector $Z^i$ generated by local trajectory encoder informatively consistent with global context $C^i$, we propose to maximize the mutual information between $Z^i$ and $C^i$ conditioned on the agent $i$'s local trajectory $B^i$. We draw the idea from variational inference (Alemi et al., 2016) and derive a lower bound of this mutual information term.

**Theorem A.1.** *Let $I(Z^i; C^i|B^i)$ be the conditional mutual information between the local teamwork situation embedding $Z^i$ of agent i and global embedding $C^i$. Then, the lower bound is given by:*

$$I(Z^i; C^i|B^i) \geq \mathbb{E}_{Z^i, C^i, B^i}\left[\log \frac{q_\xi(Z^i|C^i, B^i)}{p(Z^i|B^i)}\right]. \tag{9}$$

The lower bound can be rewritten as a loss function:

$$\mathcal{L}_{MI} = \mathbb{E}_{Z^i, C^i, \tau^i \sim \mathcal{D}}[D_{KL}[p(Z^i|B^i)||q_\xi(Z^i|C^i, B^i)]], \tag{10}$$

where $\mathcal{D}$ is the replay buffer.

*Proof.*

$$
\begin{aligned}
I(Z^i; C^i|B^i) &= \mathbb{E}_{Z^i, C^i, B^i}\left[\log \frac{q_\xi(Z^i|C^i, B^i)}{p(Z^i|B^i)}\right] \\
&= \mathbb{E}_{Z^i, C^i, B^i}\left[\log \frac{q_\xi(Z^i|C^i, B^i)}{p(Z^i|B^i)}\right] \\
&\quad + \mathbb{E}_{C^i, B^i}[D_{KL}p(Z^i|C^i, B^i)||q_\xi(Z^i|C^i, B^i))] \\
&\geq \mathbb{E}_{Z^i, C^i, B^i}\left[\log \frac{q_\xi(Z^i|C^i, B^i)}{p(Z^i|B^i)}\right],
\end{aligned} \tag{11}
$$

where the last inequality holds because of the non-negativity of the KL divergence. To prevent degenerate solutions, we employ the information bottleneck regularization, that is we add a small penalty $\mathcal{H}(Z^i|B^i)$ to maximize the entropy of $Z^i$, preventing deterministic collapse. Then it follows that:

$$
\begin{aligned}
&\mathbb{E}_{Z^i, C^i, B^i}\left[\log \frac{q_\xi(Z^i|C^i, B^i)}{p(Z^i|B^i)}\right] \\
&= \mathbb{E}_{Z^i, C^i, B^i}[\log q_\xi(Z^i|C^i, B^i)] - \mathbb{E}_{Z^i, B^i}[\log p(Z^i|B^i)] \\
&= \mathbb{E}_{Z^i, C^i, B^i}[\log q_\xi(Z^i|C^i, B^i)] + \mathbb{E}_{B^i}[\mathcal{H}(Z^i|B^i)] \\
&= \mathbb{E}_{C^i, B^i}\left[\int p(Z^i|C^i, B^i)\log q_\xi(Z^i|C^i, B^i)dZ_t^i\right] + \mathcal{H}(Z^i|B^i)
\end{aligned} \tag{12}
$$

$\square$

To illustrate the flow of Open-OTAF, we show the training procedure in Algorithm 1. A set of teammates can be generated via heuristic and RL algorithms, and we store the small batch of trajectories into a replay buffer D. The encoder and decoder are trained to force the learned representation to precisely capture the behavioral information and precisely estimate the predictive probability (Line 6). Afterward, the CRP prior and predictive likelihood are calculated to determine the assignment of the teammates $m$ (Line 7-9). Then, we update the existing cluster or instantiate a new cluster based on the assignment (Line 11-14). In the training, we first sample a teammate from the cluster and fix it in this episode. The teammates pair with the controllable agents,and they make decisions together. To train the agent policy networks and the context encoders, the moving average values of context vectors are updated, and the optimization objectives are calculated (Line 24-29).

## A.2 Details about Environment setup and Baselines

### A.2.1 Environment setup

**Setup for LBF**: In LBF, the learner retrieves objects that are positioned in an $8 \times 8$ grid world, as shown in Figure A.2(a). In the Open Dec-POMDP setting, one agent is controllable and will stay in the environment for the whole episode. The maximum number of teammates is 5, where the teammates can enter and leave the environment freely and their policy will change as well. Each agent, including the learner and their

---

**Algorithm 1** Training Procedure

---

**Require:** Batch of training teammates' behavioural policies $\{\pi_j^{-i}\}_{k=1}^K$; learning rate $\alpha$; concentration param $\lambda$, initial clustering number $M = 1$.

1: Initialize the replay buffer D
2: **while** *env* is not done **do** // `Generate data`
3:     **for** $k = 1, \ldots, K$ **do**
4:         Sample data $D^k = \{(s_t, a_t^i, \boldsymbol{a}_t^{-i}, r_t)\}_{t=1,\cdots,T}$ using $\pi^i$ and $\boldsymbol{\pi}^{-i}$.
5:         Sample small batch of trajectories $\tau^k$.
6:         Update $E_{\omega_1}, D_{\omega_2}$ by minimizing $\mathcal{L}_C$.
7:         Extract the feature of the trajectory $\tau^k$ by $E_{\omega_1}(\tau^k)$.
8:         Calculate the *CRP* prior probability $P(v_m^k)$.
9:         Calculate the predictive likelihood $P(\tau_s^k | \tau_s^k; v_m^k)$.
10:        $m* = \arg\max_m P(v_m^k) P(\tau_a^k | \tau_s^k; v_m^k)$.
11:        **if** $m* \leq M$ **then**
12:           Assign the $k^{th}$ teammate to the $m*$ cluster.
13:           Update the cluster center $\bar{v}_{m*} = \frac{n_{m*}\bar{v}_{m*} + v^k}{n_{m*}+1}$.
14:           Update the number of cluster $m$: $n_{m*} = n_{m*} + 1$.
15:        **else if**
16:           **then**Initialize the $(M + 1)^{th}$ cluster with the $k^{th}$ teammate.
17:           Initialize the cluster center $\bar{v}^{M+1} = v^k$.
18:           Update $M = M + 1$ and $n_{M+1} = 1$.
19:        **end if**
20:        $k \leftarrow k + 1$
21:     **end for**
22:     Add $\{K, P(v_m^k)\}$, $D^k$ into $\mathcal{D}$.
23:     **for** steps in training steps **do**
24:         Sample one trajectory $D \sim \mathcal{D}$
25:         **for** $t = 1, \cdots, T - 1$ **do**
26:           Compute $(\mu_{c_t^i}, \sigma_{c_t^i}) = f(s_t, \boldsymbol{a}_t^{-i}))$ and sample $c_t^i$.
27:           Compute $(\mu_{z_t^i}, \sigma_{z_t^i}) = f^*(B_t^i)$ and sample $z_t^i$.
28:           Compute $C_t^i, Z_t^i$ using HAN.
29:           Computer $\mathcal{L}_Q, \mathcal{L}_{MI}$ and $\mathcal{L}_C$.
30:           Update $G$ and global model by minimizing $\mathcal{L}_Q$.
31:           Update local model by minimizing $\mathcal{L}_{MI}$.
32:        **end for**
33:     **end for**
34: **end while**

---

teammates, is assigned a numerical level, as are the objects themselves. The level of agents and the foods are randomly chosen from $\{1, 2, 3\}$ at the beginning of an episode. In the partially observable version, agents can only observe entities in a $5 \times 5$ grid surrounding them, including the positions and levels of all the entities. Any agent has six actions, i.e., [up, down, left, right, load, no-op], where the first four actions mean the agent moves towards the corresponding direction, load means loading food next to it, and a no-op action indicates doing-nothing during an entire episode. The agents' objective is to collect all the objects and the collection is only successful if the sum of the involved agents' levels is equal to or greater than the item level. Every agent that collects an object is given a reward equal to the level of the object. An episode terminates if all available objects are collected or after 50 timesteps.

**Setup for Wolfpack**: In Figure A.2(b), the multiple hunters attempt to catch randomly prey in a grid world of size a $10 \times 10$, where the "catching" means the prey is in the cardinal direction of at least two hunters. Partial observability is induced by limiting the learner's observation to entities within a Manhattan distance of 3 from itself. The observation of each agent is the coordinates of its location, its ID, and the coordinates of the prey $k$ if observed. Agents in this environment have five actions, i.e. [up, down, left, right, no-op]. Every hunter in a pack that captured prey is given a reward of two times the size of the capturing pack. We penalize agents by $-0.5$ for positioning themselves next to prey without teammates positioned in other adjacent grids from the prey.

**Setup for Penalized Cooperative Navigation (PCN)**: In PCN, multiple players must cooperate through physical actions to reach a set of landmarks as shown in Figure A.2(c). The agents need to learn to infer the landmark they must cover, and move there while avoiding other agents. The action space of each agent

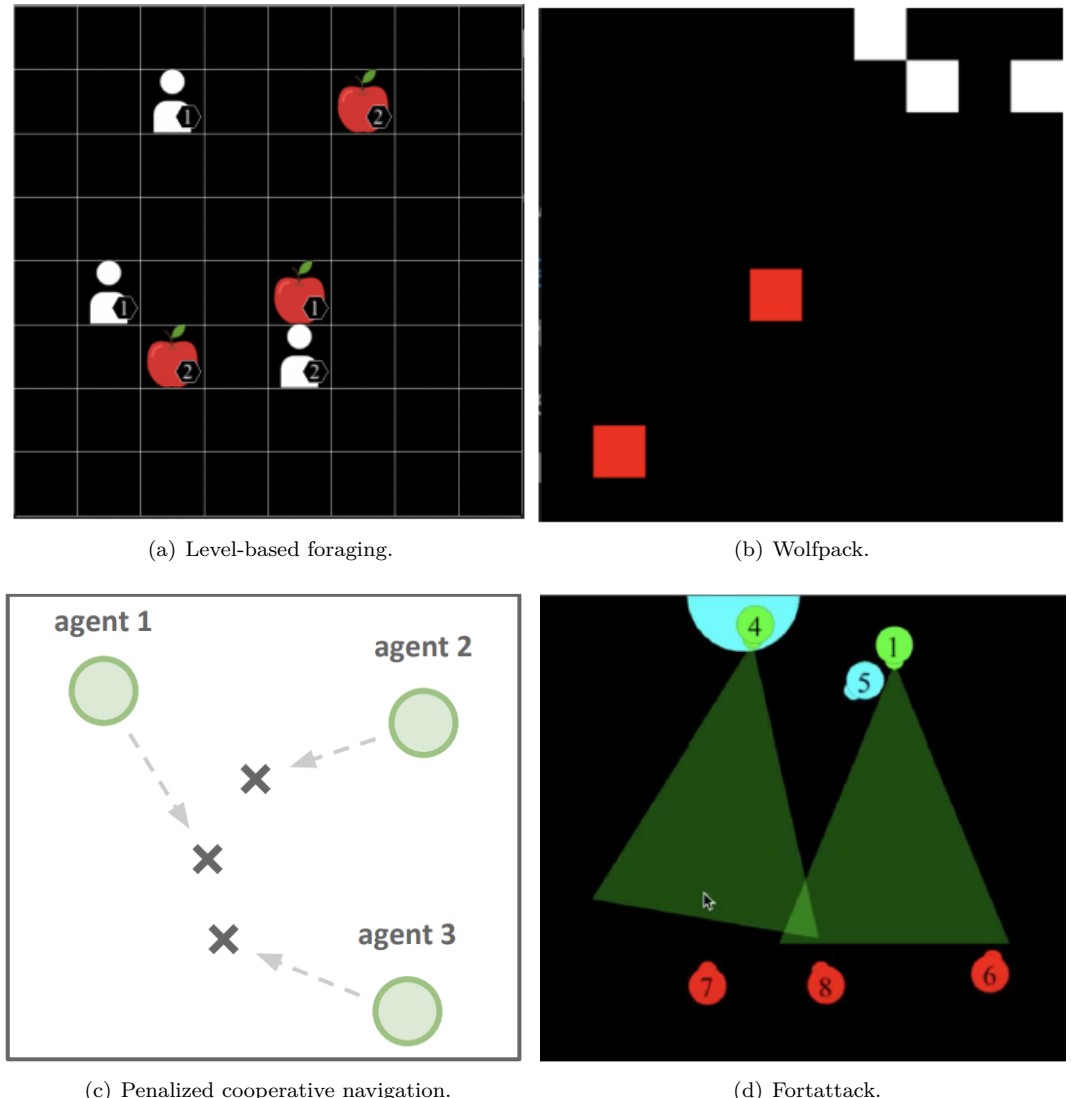

(a) Level-based foraging.

(b) Wolfpack.

(c) Penalized cooperative navigation.

(d) Fortattack.

Figure A.2: State visualization of benchmark environments.

contains five discrete movement actions, i.e. [up, down, left, right, no-op]. Their objective is to simultaneously cover two destination grids to get a reward of 1. However, a penalty of $-0.2$ is imposed on the learner if it reaches a destination without the presence of teammates simultaneously covering the other destination. Similar to LBF, the learner can only see the destination grids or teammates if they are inside a $5 \times 5$ region surrounding the learner.

**Setup for Fortattack:** The FortAttack environment defines a bounded two-dimensional space where agents are constrained within specific coordinate ranges. In Figure A.2 (d), the horizontal axis is restricted to values between $-0.8$ and $0.8$, while the vertical axis spans from $-1$ to $1$. The fort is modeled as a semicircle with a center at $(0.0, 1.0)$ and a radius of $0.3$. During initialization, the attackers are positioned with their vertical coordinates set between $-1$ and $-0.8$, ensuring they begin at a distance from the fort. This positioning allows the attackers to start from a location far enough to strategize their approach. Conversely, the defender's initial position is constrained to the vertical coordinate range between $0.8$ and $1$, placing them closer to the fort, thus providing an advantageous starting position for defense.

### A.2.2 Teammate Types

In open ad hoc teamwork, the primary challenge lies in the fact that the controlled agent lacks prior knowledge of its prospective teammate to interact with. Hence, we randomly choose multiple teammate types to contemplate all potentialities in simulating an open ad hoc teamwork environment where an agent may need to cooperate with unfamiliar counterparts. Therefore, in all environments, each type of teammate policy is implemented either via different heuristics or reinforcement learning-based policies. We vary the teammate's policies in terms of their efficiency in executing a task and their roles in a team. To create a diverse set of teammates for open ad hoc teamwork, we used the following mixture of heuristics along with RL-based models to control teammates during the training process. Our method aims to learn teamwork situations to cooperate with previously unknown teammates' behaviours. To investigate the policy transfer ability and generalization of the proposed methods, we evaluate the method by comparing the performance under different teammate types.

For each environment, we design 20 different policies, and we have one of 10 policies as the training set and the other 10 policies as the testing set. Specifically, in the Wolfpack environment, we used the 9 heuristic policies proposed by (Barrett et al., 2011) along with an A2C algorithm (Mnih et al., 2015) to control teammates. Similar to Wolfpack, we also create a diverse set of teammate types for LBF which requires agents to adapt their policies towards their teammates to achieve optimal performance. With LBF, we use 8 heuristics (Albrecht & Stone, 2019; Albrecht & Ramamoorthy, 2015), an A2C algorithm (Mnih et al., 2015) and DQN (Mnih et al., 2016) as our teammate policies. During the testing, for Wolfpack and LBF, we use 10 RL-based policies, which were implemented by 5 different MARL algorithms to develop several teams of agents. To ensure diversity, we use different random seeds for each RL-based algorithm and save the corresponding models at 2 different checkpoints. On the other hand, for PCN, we utilize 5 different MARL algorithms, to develop 15 different policies showing distinct policy representations from all developed models. In the training, we adopt 5 heuristics (Rahman et al., 2022) and random sample 5 different policies to control the teammates. The other 10 teammate policies as the testing set. In Fortattack environments, the policies governing each teammate type are implemented using either 10 distinct behavioral heuristics or MARL-based approaches.

### A.2.3 Environment Openness

We create an open process that determines how agents enter and leave during episodes. The number of timesteps an agent can exist within the environment is determined by sampling from a certain range of integers. After staying for the predetermined number of timesteps, agents are removed from the environment. After that, agents can reenter the environment after a specific period of waiting time. The type of an agent entering an environment is uniformly sampled from all available types.

In all environments, a teammate only exists in the environment for a certain number of timesteps. If a teammate has surpassed its designated temporal lifespan, it is immediately removed from the environment. A teammate that has been removed is allocated a waiting period, which is the duration before it is pushed into a reentry queue. Given a non-empty reentry queue, agents within the queue are reintroduced to the environment if the number of agents does not exceed the aforementioned upper limit. It is important to note that the reentry queue is randomized, thus inducing an aleatory team composition during learning. The maximum number of players for training is 3 and the maximum number of players for training is 5. For Wolfpack, teammates' lifetime is sampled uniformly between 25 and 35 timesteps while the waiting period is sampled uniformly between 15 and 25 timesteps. By contrast, in the LBF, PCN environment, the teammates' lifetime is sampled uniformly between 15 to 25 timesteps while the waiting period is sampled uniformly between 10 and 20 timesteps.

Unlike LBF and Wolfpack, where agent counts remain fixed, FortAttack dynamically adjusts the number of agents based on in-game actions: an agent is removed only when shot by an opponent. The respawn delay is determined by the distance to the shooter at the time of elimination, following a linear interpolation—agents shot at the closest possible distance (minimum separation) incur an 80-timestep penalty, while those hit at longer distances experience proportionally shorter delays. This design ensures that tactical positioning

Table A.2: The comparison between different algorithms.

| Method | GNN | VAE | AM | MI | CRP | HAN |
|---|---|---|---|---|---|---|
| MADDPG | | | | | | |
| GPL | √ | | √ | | | |
| VAE-GPL | √ | √ | √ | | | |
| ODTIS | √ | √ | | √ | | |
| Open-OTAF(GNN) | √ | √ | √ | √ | √ | |
| Open-OTAF | | √ | √ | √ | √ | √ |

Table A.3: Hyperparameters in the experiments

| Notation | Meaning |
|---|---|
| $T$ | 640,0000 |
| $\alpha$ | 1 |
| $\gamma$ | 0.99 |
| Optimizer | Adam |
| Batch size | 128 |
| Learning rate | $1e-4$ |
| Evaluation interval | 3200 |
| Action selector | $\epsilon$ greedy |
| $\epsilon$ start | 1 |
| $\epsilon$ finish | 0.05 |
| Replay memory size | 5000 |

influences respawn timing, adding strategic depth. At episode initialization, agents are evenly distributed between attacking and defending teams, with counts set to the predefined maximum.

### A.2.4 Baselines

MADDPG is a benchmark of MARL algorithm for closed environments. We employ MAPPDG to control the ad hoc agents, using the learner's observations as input to the policy. GPL and VAE-GPL are two type-based methods designed for addressing open ad hoc teamwork problems. Specifically, GPL is suited for scenarios with full observability, while VAE-GPL is applicable in settings with partial observability. Although GPL assumes full observability of the state, we apply GPL in our experiments by treating the learner's observations as input. ODITS improves zero-shot coordination performance in an end-to-end fashion. Two variational encoders are adopted to improve the coordination capability. As ODITS considers only a single ad hoc agent in the closed environment, we further extend the ODTIS to open environments by introducing a graph neural network. Moreover, to demonstrate the advantages of HAN, we also include a comparison with the Open-OTAF(GNN) algorithm, where GNNs are employed to produce the representations of teamwork situations.

### A.3 Algorithm configuration

For the LBF, wolfpack, PCN and Fortattack, the training steps $T$ are 640, 0000 and we evaluate all the methods with 100 test episodes after every 3200 training steps. The parameters of the networks are updated by `Adam` optimizer (Kingma & Ba, 2015) with a learning rate $1e-4$ for all environments. The discounting factor is $\gamma = 0.99$, learning rate $\alpha$ and the concentration hyperparameter is $\lambda = 1$. Since they induce the best performance compared with other values. For exploration, we use $\epsilon$ greedy from 1.0 to 0.05. Batches of 128 episodes are sampled from the replay buffer, and all components in the framework are trained together in an end-to-end fashion. More details of the hyper-parameters are provided in Table A.3. All experiments are carried out in a machine with Intel Core i9-10940X CPU and a single Nvidia GeForce 2080Ti GPU.

In the CRP-based teammate cluster model, we introduce an encoder $E_{\omega_1}$ to model the teammate trajectory into a latent space as their behaviour type, where $E_{\omega_1}$ is implemented by a long short-term memory (LSTM) network which takes $\tau_t$ as inputs and outputs 16-dimensional behavioural embeddings $v$. To predict the likelihood of the action, we use an RNN-based decoder $D_{\omega_2}$ that consists of a GRU cell whose hidden dimension is 16, takes $\tau_t^s$ and $v$ as input and reconstructs the action $a_t$. we set the maximum of cluster number as 10 in all environments.

In the global teamwork situation inference model, the latent variable inference network is implemented as a 2-layer MLP and LSTM which receives the global trajectory as input. More details are provided in Figure A.2. It subsequently produces the mean and covariance matrix for the variational parametric distribution. Assuming that $h_t$ and $e_t$ are the hidden and cell states of the LSTM at timestep $t$, the LSTM updates the vectors following:

$$\mu_{c_t^i} = MLP_{\alpha^\mu}(e_t), \sigma_{c_t^i} = MLP_{\alpha^\sigma}(e_t), \tag{13}$$

where $e_t, h_t = LSTM_\alpha(H_t, e_{t-1}, h_{t-1})$ and $H_t = [s_t, a_t^{-i}, \{k, p(v_k^m | \tau_k)\}]$. The latent variable $c_t$ vector will be sampled from the distribution $c_t \sim \mathcal{N}(\mu_{c_t}, \sigma_{c_t})$. Then we use the inherent ability of graph neural network to transform the dynamic-size team's information into a fixed-size output. As for the local teamwork situation inference model, it receives the ad hoc agent's preprocessed observation as input and outputs $z_t$.

To capture the heterogeneity among teammates and handle dynamic team sizes, we model the team as a heterogeneous graph and introduce a HAN to compute the final teamwork situation embedding. The global teamwork situation embedding $c_t^i$ and predicted teammate cluster information will be concatenated as the node input of the HAN to calculate the fixed embedding $C_t^i$. To estimate the joint value function, we first map $C_i^t$ into the parameters of $G$ by a hyper-network. Then, $G$ maps the ad hoc agent's utility $u_i$ into the value estimation. This alternative design changes the procedure of information integration.

For the ad hoc agent, we utilize coordination graphs and agent modelling to calculate the marginal utility by considering the effect of other agents as shown in equation 7 (in the main paper). More details are provided in Figure A.4. The single utility of $Q_\pi^i(a_t^i | Z_t^i)$ and pairwise utility $Q_\pi^{i,j}(a_t^i | Z_t^i)$ are implemented as multilayer perceptions (MLPs) parameterised by $\beta$ and $\delta$ respectively. Concretely, given $Z_t^i$, the agent modelling module computes the likelihood of observed agents' actions $q$ by multilayer perceptrons (MLPs). The agent marginal utility can be calculated by equation 7. After that, the marginal utility $\mu_t^i$ will be fed into the mixing network to calculate the joint action function finally. To maximize the mutual information between local and global teamwork situations embedding conditioned on the agent $i$'s local trajectory, a variational distribution network, which is similar to the global inference network, is used to approximate the conditional distribution.

### A.4 Experiment Results

To determine the importance of each component of Open-OTAF, we also conducted an ablation study of the different components under different scenarios. The comparison results under Wolfpack, PCN and Fortattack environment are provided in Figure A.4. The results show that the $w/o$ MI variant exhibits the most significant performance degradation across all games, highlighting the importance of incorporating global information to support the learning of the local teamwork situation inference model. Additionally, $w/o$ CRP performs worse than Open-ATAF, indicating that the CRP-based teammate model enables the controlled agent to adapt more effectively to dynamic teammates. Moreover, we observe that agent modeling contributes to learning more accurate marginal utilities by accounting for the influence of other teammates, leading to modest improvements in coordination.

The testing results are also provided in Figure A.5, where we provide the average return (bars) and the standard deviation (error bars) over different seeds. The results show that w/o CRP and w/o MI suffer the most severe performance degradation in all environments when compared with Open-OTAF. It shows that the introduction of the CRP model for teammate types classification and the guidance of global information help agents to estimate more accurate teamwork situations.

The cluster number m may influence the algorithm's performance. We conduct an ablation study for different values of cluster number under different scenarios to further reveal the influence of the cluster number during

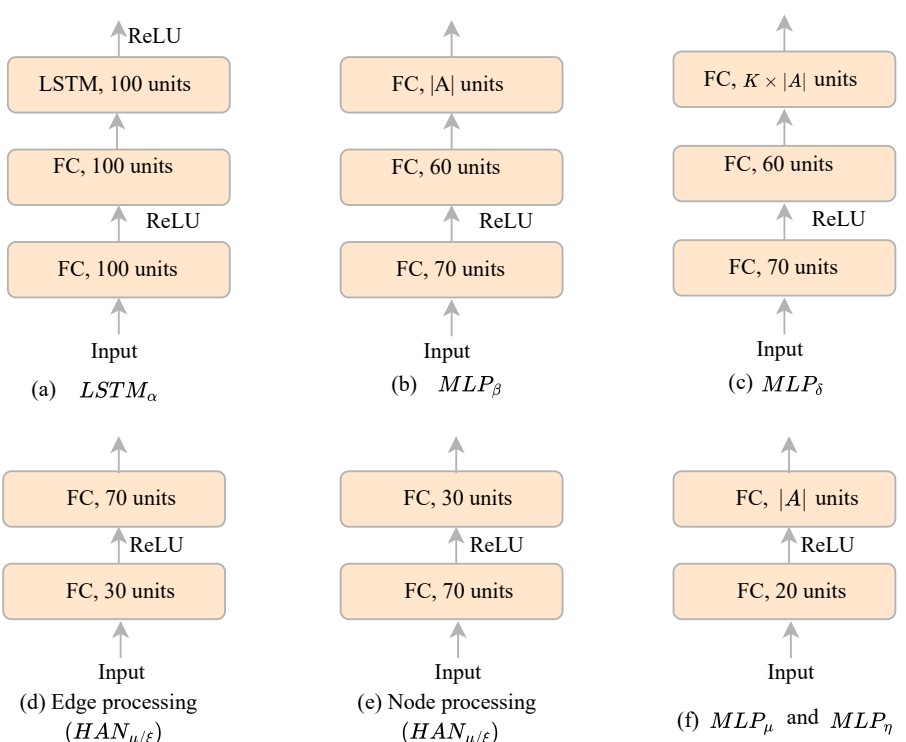

Figure A.2: Architecture details of teamwork situation inference model: (a) the architecture used in teamwork situation inference network; (b) singular utility computation; (c) pairwise utility computation; (d) edge embedding computation in the agent and auxiliary agent model; (e) node embedding computation in the agent and teammates; (f) the MLP used by the agent and teammates to process the resulting GNN node embeddings. FC denotes a fully connected layer, LSTM denotes an LSTM layer, and the accompanying number denotes the size of the layer. Labels on the arrows indicate the non-linear functions used between the layers while no labels indicate no non-linear functions being applied to the resulting output vectors.

training. We vary the cluster number to 5, 8, 10, 12, and 15, respectively, while fixing other configurations. Here, we present the evaluation results of our method under different scenarios (Table A.4). We can see that the performance of $m \geq 8$ is consistently better than that of $m = 5$. This observation aligns with the intuition that well-separated clusters enhance the inference of teamwork situations by more effectively capturing similarities among teammates, thereby improving overall performance. Experimental results confirm that performance increases with the number of clusters. However, it is noteworthy that the improvement exhibits only marginal gains when the cluster count exceeds 10, suggesting a saturation effect in the benefits of further granularity. To balance computational overhead and performance, we set $m = 10$ throughout the experiments.

Table A.4: The Evaluation Results for our methods on various cluster numbers $m$.

| Env | m=5 | m=8 | m=10 | m=12 | m=15 |
|---|---|---|---|---|---|
| LBF | 2.56 | 2.62 | **2.77** | 2.78 | 2.77 |
| Wolf | 21.56 | 26.98 | **27.19** | 27.21 | 27.26 |
| PCN | 0.81 | 0.83 | **0.91** | 0.93 | 0.95 |
| Fortattack | 0.57 | 0.59 | **0.68** | 0.69 | 0.69 |

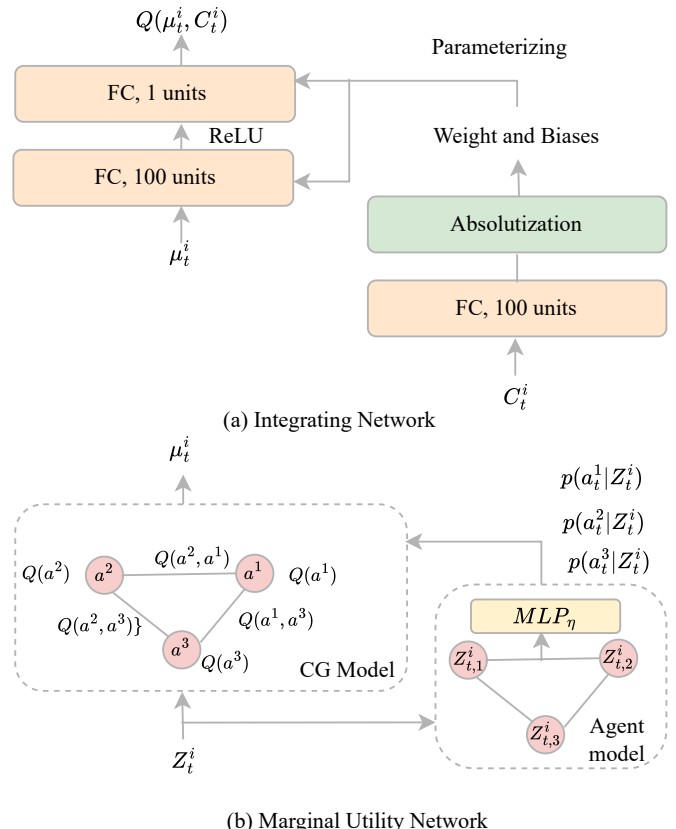

(a) Integrating Network

(b) Marginal Utility Network

Figure A.4: Architecture details of interacting network and marginal utility network.

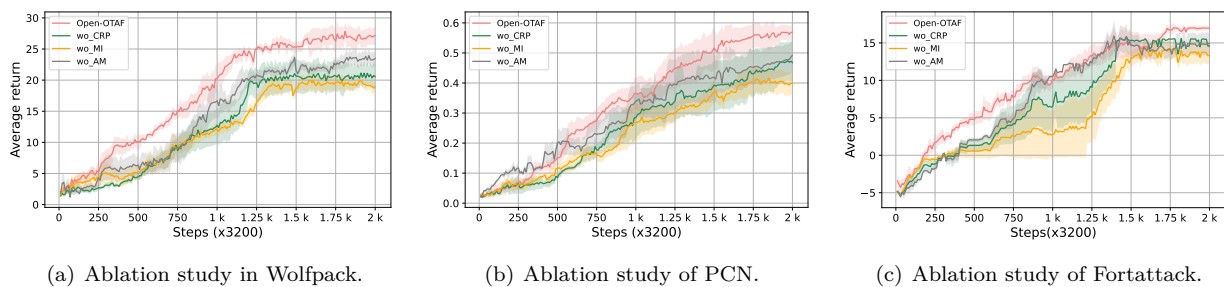

(a) Ablation study in Wolfpack.

(b) Ablation study of PCN.

(c) Ablation study of Fortattack.

Figure A.4: Ablations for different components under Wolfpack, PCN and Fortattack environment.

Intuitively speaking, adapting with dynamic teammates with local information seems more complicated and may need more data. To investigate whether the HAN, CRP, and AM can increase the computation complexity in our method, we provide the computational cost between different algorithms. We gather the number of floating-point operations (FLOPS) of a single inference and the number of parameters for each method. Table A.5 shows the computation complexity of different algorithms. It shows that our method has a comparable model volume and computational complexity as baselines. This similarity arises because all methods

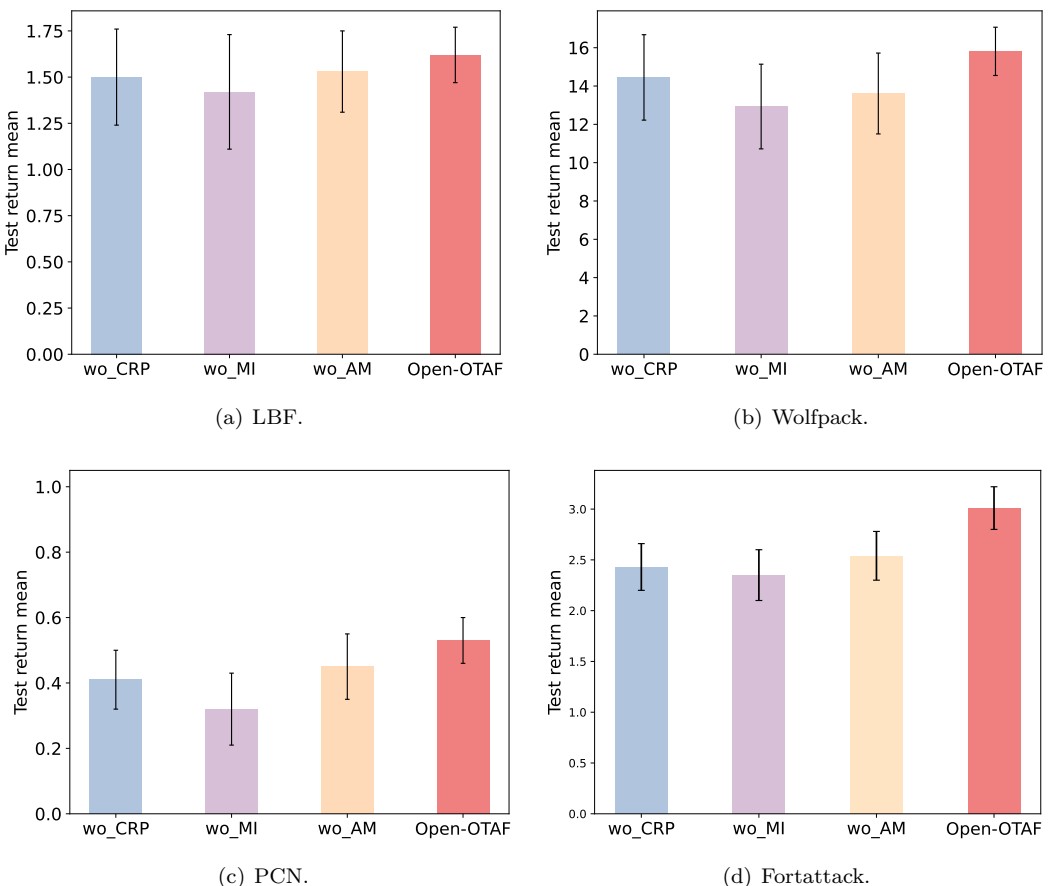

Figure A.5: Testing performance comparison across various scenarios for different components.

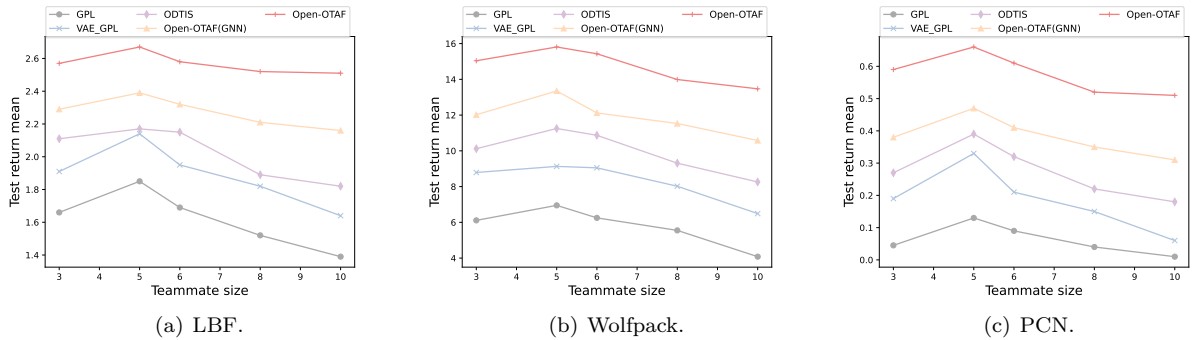

Figure A.6: Generalization study with varying teammate size.

share the same CTDE framework, whose primary bottleneck is centralized training. CTDE's cost stems from processing high-dimensional joint observation-action spaces and scales poorly with team size (curse of dimensionality). It requires agents to learn coordination strategies while simulating teammates' behaviors, alongside iterative gradient synchronization and policy updates. While CTDE enhances collaboration, these demands slow training, especially in partially observable environments.

Table A.5: Computation cost under different methods

| Methods | LBF | | Wolf | | Fortattack | | PCN | |
|---|---|---|---|---|---|---|---|---|
| | Params(M) | FLOPs(G) | Params(M) | FLOPs(G) | Params(M) | FLOPs(G) | Params(M) | FLOPs(G) |
| MADDPG | 7.98 | 0.100 | 8.09 | 0.102 | 8.03 | 0.101 | 7.89 | 0.102 |
| GPL | 8.01 | 0.102 | 8.11 | 0.104 | 8.07 | 0.104 | 7.96 | 0.106 |
| ODTIS | 7.99 | 0.101 | 8.10 | 0.102 | 8.05 | 0.103 | 7.95 | 0.105 |
| Open-OTAF | 8.02 | 0.104 | 8.13 | 0.105 | 8.08 | 0.106 | 7.98 | 0.107 |

