# OpenReview forum: "Collaboration with Dynamic Open Ad Hoc Team via Team State Modelling"
_TMLR — Accepted by TMLR_

### Review · Reviewer_69MY · 2025-04-27

**Summary Of Contributions:**

The manuscript introduces Open-OTAF (Open Online Teammate Adaptation Framework), which addresses the problem of open ad hoc teamwork under partial observability and dynamic settings. Collaboration here includes interactions with possibly unknown, heterogeneous teammates whose number can dynamically change over time of execution. Differently from previous research, the presented framework does not make limiting assumptions on team composition and allows decentralized execution.

The presented framework addresses online adaptation with varying numbers of heterogeneous teammates in open ad hoc teamwork setting by proposing 3 main components:
- A Chinese Restaurant Process (CRP)-based clustering approach to accommodate different teammate behaviors, categorizing diverse teammate policies into distinct clusters to reduce the complexity of the context search space.
- A heterogeneous graph attention network (HAN) adopted to learn effective teamwork situation representations that accommodate variable team sizes and diverse inter-agent relationships.
- A conditional policy that enables the ad hoc agent to adapt dynamically to new teammates' behaviors by conditioning its policy on the locally inferred teamwork situation.

Experimental validation across four multi-agent benchmarks (Level-Based Foraging, Wolfpack, Penalized Cooperative Navigation, and FortAttack) demonstrates superior performance compared to baselines (MADDPG, GPL, VAE-GPL, ODTIS), with comprehensive ablation studies confirming the necessity and effectiveness of each key component. The empirical results indicate not only higher average returns but also significantly faster convergence speed.

**Audience:**

Yes

**Claims And Evidence:**

Yes

**Requested Changes:**

Critical points:

The notation adopted across the manuscript and specified in Table 1 lacks consistency, which significantly impedes comprehension. Some points that must be addressed includes:
- Inconsistent formatting style for mathematical entities e.g., bold font adopted for agent sets appears inconsistently, and where curly braces are used to represent sets clear index bounds should be defined
- Switching between superscript and subscript index position in occurrences of the same variables e.g., $o$, $a$, $\beta^{\Phi^i}$, $C^i_t$ in Sec. 3, 4, A, Algorithm 1 and Fig.1, 3
- Inconsistent meaning of sub/superscripts throughout the paper. A systematic approach should be adopted where, for example, superscripts consistently denote agent indices while subscripts indicate temporal indices.
- Variable definitions changing between main text and supplementary material e.g., concentration parameter symbol differs between Section 4.2 and Algorithm 1/A.3

Aditionally, the notation results excessively complex and specific, leading to multiple symbol overlaps and definitional ambiguities. I recommend simplifying the notation through several approaches (not mandatory suggestions):
 - $\omega = ${$ \omega_1, \omega_2$} parameter set could easily be simplified to a single larger parameter set $\omega$
 - Remove variables that are introduced but never utilized
 - Omit time subscripts in explanatory sections where temporal indexing is not essential (with appropriate notation in the text)
 - Consider standardizing agent indexing to use either $i$ or $k$ consistently

Specific examples of problematic notation from Table 1 include:
- $T_k$ indicates $k^{th}$ teammate and $T$ the time horizon
- $D_{\omega_2}$ represents the trajectory decoder while $D$ denotes sampled data from replay buffer
- P (number of meta-paths) is introduced but never used
- $\mathcal{N}$represents both the dynamic set of agents and the normal distribution

---
Major points (important clarifications needed and strongly suggested changes):

- Sec.2: The statement "The policy gradient methods are a sub-class of CTDE" is erroneous unless restricted to Cooperative MARL contexts. Please rephrase.
- Fig.1 and Fig.3: Font size and element dimensions results too small for comfortable reading.
- Sec.3: The variable $Q$ is used without formal introduction as joint action value function.
- Sec.4.2: The explanation of variable $v^m_k$ requires substantial clarification. The text states "$v^m_k$ represents that the $k^{th}$ teammate belongs to the $m^{th}$ cluster" suggesting a boolean assignment variable, which aligns with the prior distribution definition. However, in the predictive likelihood equation, computed by the decoder, $P(\tau^{\alpha}_{k} | v^m_k; \tau^s_k) = D(\tau^{\alpha}_k | \tau^s_k ; v^m_k)$, following A.3 (in the network input description) and Eq.1, the Decoder computing the predictive likelihood appears to be conditioned on the Encoder's output features i.e. $v_k$ set of features rather than the $v^m_k$ boolean variable. Additionally, the cluster assignment process "we are able to decide which cluster the $k^{th}$ teammate belongs to" lacks formal definition in the main text (I assume $m = argmax_m P(v^m_k|\tau_k)$ from Algorithm 1); $\bar{v}^m$ has not been formally defined and neither its update following teammate cluster assignment $\bar{v}^m = \frac{n^m \bar{v}^m + v_k}{n^m+1}$ when $m \leq M$, else $\bar{v}^m = v_k$ if $m = M +1$ (from Algorithm 1).
- Eq.1: Please clarify whether the negative log likelihood loss is computed on the set of trajectory actions, as per equation definition, or only on the last action $a_t$ as described in A.3.
- Sec.4.3: The sentence "Note that the size of {$c^t_k$} corresponds to the number of teammates, which can change across episodes in the open ad hoc teamwork" raises questions about whether {$c^k_t$} and {$k$, $p(v^m_k|\tau_k)$} have sizes equal to or upper-bounded by the number of teammates (changing across the episode). Additionally, the section's opening statement "After the teammates are divided into different clusters, we randomly choose multiple teammates to interact with the controlled agent" suggests that the teammates are clustered and then a number of those (fixed number?) is randomly sampled and considered in the global teamwork situation computation. Please clarify this point and also if the sampling takes in consideration the clustering subdivision.
- Sec.4.3: Is it correct to state that the two meta-path graphs are complete graphs that are learned in the sense that the node-level attention weights the connections between adjacent nodes? I have some doubts regarding this characterization. Alternatively, do the graphs representing coordination and no-coordination reflect fixed relationships that are part of the adopted environment? Additionally, please clarify the definition of neighbors in the graph: does the notation $\mathcal{N}$ refer to a specific subset of the entire dynamic set of agents involved in the task? Lastly, how are different meta-paths able to learn distinct relationships, and where is evidence of this presented in the work?
- Sec.4.3: More details about $att_{node}$ and the semantic level attention network are required as their structure and descriptions are not provided. No further detail can be inferred from section A.3 where HAN network is cited (also Fig.A.2 d and e) but without explicit description of such components.
- Sec.4.3: The rationale for adopting the hypernetwork $G$ is inadequately discussed.
- Sec.4.4: Local trajectory $B$ has been mentioned without formal definition of what it represents and how $\mu_{\phi_i(B^i_t)}$ and $\sigma_{\phi_i(B^i_t)}$ have been obtained and used in place of $\mu_{c_t}$ and $\sigma_{c_t}$.
- Sec.5.2.1: The text states "These results show that our method can design an autonomous agent capable of robust teamwork under dynamic teammates without pre-coordination" however the dynamic aspects involved in the experimental setup are not explained in the main text, but reported only in the supplementary material.
- Fig.6: The ablation graph does not specify which environment was used for testing.
- A.2.1: The "Setup for Fortattack" paragraph lacks a description of the game dynamics.
- A.2.2 and A.2.3: Details about the Fortattack environment are missing.
- Fig.A.2: Please clarify what is meant by "auxiliary agent model".

----
Minor points (e.g. typo, personal suggestions):

- Sec.1: Consider adding citations to communication/message passing methods literature to strengthen the claim "achieving online adaptation with dynamic teammates in open AHT environments remains an open challenge". In such research branch, heterogeneous collaboration in a POMDP setting is a topic with relevant works but faced with additional cost of information sharing to develop beliefs of other agents.
- Sec.1: "Section 2 provides a briefly review" -> "brief"
- Sec.2: "These works [Barrett et al.(2017)]" is poorly phrased
- Sec.2: "Agent modellingg" -> "Agent modelling"
- Sec.4.1: "notifications" -> "notations"
- Sec.4.1: "[...]. Then, we can then" is redundant
- Sec.4.2: "demotes" --> "denotes"
- Sec.4.2: Add reference to derivation in A.1 when presenting the conditional probability proportional relation
- Sec 4.3: "g denote the activate function" -> "activation" ?
- Sec 5.1: "the environments and aldorithm configuration are provide in" -> "algorithm", "provided"
- Sec 5.2.2: "The results that the" is poorly phrased
- Sec 5.2.3: "methods that achieve low returns" -> "high" ?
- A.2.2: "[...]. Specificity, in the Wolfpack environment" -> "Specifically"
- A.2.2: "5 different MARL-based to develop" -> "MARL algorithms" ?

**Strengths And Weaknesses:**

The manuscript presents several notable technical contributions:
- The paper establishes a rigorous mathematical relationship between marginal utility and team utility, demonstrating that $arg max_{a^i} u^i(s, a^i, \textbf{a}^{-i}) = arg max_{a^i} Q^{π_i}(s, a^i, \textbf{a}^{-i})$. This theoretical foundation justifies the optimization of marginal utility as a means to ensure optimal team performance, providing a solid basis for the proposed framework.
- The teammate clustering approach combines a Chinese Restaurant Process (CRP) for prior estimation of teammate behaviors with neural network-based likelihood estimation. This probabilistic framework enables effective categorization of diverse teammate policies into distinct clusters, thereby reducing the complexity of the context search space and facilitating more efficient representation learning.
- A significant contribution is the employment of a Heterogeneous Graph Attention Network (HAN) as an alternative to conventional graph neural networks. This design choice enables the transformation of dynamic-size team information into fixed-size output embeddings, effectively addressing the variable team composition challenge intrinsic to open teamwork and aiming to capture the heterogeneous relationships among agents.
- A rigorous theoretical foundation is provided to justify the Centralized Training with Decentralized Execution (CTDE) paradigm through formal derivations. By ensuring that local teamwork situation embeddings maintain informative consistency with global context, the framework enables an ad hoc agent to derive proxy representations of the teamwork situation solely from local observations. This theoretical basis establishes the validity of the decentralized execution mechanism.

The manuscript exhibits significant deficiencies in its presentation quality that substantially impair comprehension and verification of the technical contributions. The inconsistent mathematical notation throughout the paper creates ambiguity regarding the precise mechanisms being proposed. Furthermore, the nomenclature used to describe key components varies across sections introducing confusion. Critical procedural details of the framework implementation are inadequately specified by lack of clarity or defined in the supplementary material and not addressed in the main text.

---

### Review · Reviewer_pDn3 · 2025-05-09

**Summary Of Contributions:**

The paper introduces Open-OTAF (Open Online Teammate Adaptation Framework), a novel framework designed to enable an autonomous agent to collaborate with unknown and dynamic teammates in partially observable and open environments -- settings where team composition can change over time. Specifically, the paper tackles *open ad hoc teamwork*, an important variant of multi-agent collaboration where both the number and type of teammates vary dynamically. The authors go beyond standard assumptions like fixed number of teammates and types, or full observability.

Let me just start by saying that by no means is this framework simple -- and indeed has a lot of moving parts. I had to revisit previous sections on multiple occasions to be able to keep track of the underlying story.  That said, here’s how I understand its main contributions:

* A Chinese Restaurant Process (CRP) for online clustering of teammate behaviors. This is required so as not to model every single teammate individually, but by behavior types. It also allows for incoming individuals to either form new clusters or join a previous one. This gives a clustered representation for each teammate. Since the team size is dynamic, some modeling is required to obtain a single representation of the entire teamwork. Therefore, the authors propose their next contribution, HAN.

* Heterogeneous Graph Attention Network (HAN) is then employed to map these teammate clusters into a fixed-size embedding of the current teamwork situation, accounting for variable team sizes and heterogeneous relationships between agents.

* Mutual Information (MI) Loss is used to align the agent’s local belief (formed from partial observations) with the global teamwork situation inferred during training, helping the agent generalize at test time when global information is unavailable.

Overall, the empirical results show that Open-OTAF outperforms several baselines like MADDPG and GPL, both in performance and convergence speed, across a suite of benchmark multi-agent tasks.

**Audience:**

Yes

**Claims And Evidence:**

Yes

**Requested Changes:**

Aside from addressing the questions raised above, I encourage the authors to provide more intuitive explanations throughout Section 4 -- about how each component is essential to the story. Posing questions every once in a while would help as you move through subsections within Sec 4.0.

**Strengths And Weaknesses:**

Strengths:
* **Flexible**: Can handle dynamic team sizes and allows for the discovery of novel behavior types during training.
* **Comprehensive Evaluation & Ablation studies that further isolate the contribution of each component**

Weaknesses:
1. The combination of CRP, VAE, HAN, CG, and MI loss makes the **framework conceptually dense**. The practical feasibility of deploying such a system at scale should be discussed in more detail. The paper lacks discussion on training stability, sensitivity to hyperparameters, or engineering challenges.
2. **Limited Discussion on Scalability and Efficiency**: There is no analysis of training time, runtime cost, or memory footprint, especially compared to simpler baselines. Given the added architectural complexity, such discussion would be important for real-world adoption.
3. **Sparse Justification for Design Choices**: While the paper ablates some components, it doesn't completely offer qualitative insights into why certain design choices and the readers are left to figure it out for themselves.

Questions:

1. **Section 4.2 – Cluster Mean Update:** The paper defines the candidate cluster embedding as $v_k^m = \frac{n_m \bar{v}_m}{n_m + 1}$, but this expression appears to omit the embedding of the current teammate $v_k$. Shouldn't the correct update be $v_k^m = \frac{n_m \bar{v}_m + v_k}{n_m + 1}$ to reflect the posterior mean after incorporating $v_k$?

2. What is the computational cost (training time, memory) of Open-OTAF relative to baselines like MADDPG or VAE-GPL?

3. Can you elaborate more on what meta-paths are and its overall role in the performance?

4. What prevents Equation 5 from collapsing into a degenerate solution where $\mathcal{Z}_t$ collapses to something uninformative like a constant? Aside from empirical results, of course.

---

### Review · Reviewer_B9hn · 2025-05-20

**Summary Of Contributions:**

This paper proposes Open-OTAF, a novel MARL framework that enables adaptation to dynamic teammates in open ad hoc environments by combining CRP-based teammate clustering, heterogeneous graph attention networks, and mutual information-guided local-global teamwork situation inference.

**Audience:**

Yes

**Claims And Evidence:**

No

**Requested Changes:**

See weaknesses.

**Strengths And Weaknesses:**

**Strengths**

1. The paper tackles open ad hoc teamwork with dynamic teammates, which is a more realistic setting than prior fixed teammate cooperation tasks.
2. The overall algorithmic framework is reasonably designed, with each module serving a specific purpose, and the effectiveness is supported by ablation studies.
3. The experiments across the four environments demonstrate consistently strong performance.

**Weaknesses**

1. The experimental environments are quite simplistic, confined to 2D worlds and discrete action spaces. There are concerns about whether the proposed algorithm can be effectively extended to more complex interactions.
2. The proposed components of the algorithm, such as CRP Clustering and HAN, introduce noticeable computational overhead, which may pose challenges for larger-scale experiments.
3. It is unclear how the maximum number of clusters influences the experimental results. From my understanding, the CRP Clustering is performed based on the rollout trajectories of teammate agents. However, during the course of interaction, do teammate policies remain fixed, or are they updated? Even if the teammates follow fixed policies, their behaviors may still change in response to updates in the controlled agent’s policy. Would this not lead to a proliferation of possible teammate types, potentially making clustering more difficult?

---

### Review · Reviewer_nBpW · 2025-05-20

**Summary Of Contributions:**

This paper contributes Open Online Teammate Adaptation Framework (Open-OTAF), which is designed for online adaptation in open ad-hoc teams (environments with unknown teammates), where teammates can have dynamically evolving numbers and strategies. The framework is presented for fully cooperative games where all agents are assumed to share a common goal. The agents in Open-OTAF aim to compute a "teammate situation" (an embedding) using Graph Attention Networks. This embedding forms the basis for learning the best responses for each agent. The framework uses the centralized training and decentralized execution (CTDE) paradigm, where the centralized training uses global context with sampled representative teammates and the decentralized execution simply relies on the local observations.  The functionality and superiority of Open-OTAF is empirically demonstrated on multiple MARL test beds.

**Audience:**

Yes

**Claims And Evidence:**

Yes

**Requested Changes:**

Please see "Weaknesses" above.

**Strengths And Weaknesses:**

Apologies for the delay in submitting this review. Before accepting to review this paper, I had requested the AE for additional time to review based on my other commitments. I only accepted to review this paper after the additional time was granted by the AE. I assure the authors and the AE that I have written my review independently without looking at the other submitted reviews.

Strengths:

1. The paper studies a very challenging and timely problem.

2. Experiments appear to be very comprehensive.

3. The final insights that were presented in the second paragraph of Page 12 (paragraph just before Conclusion) are very clear and comprehensive. But, I feel that this is coming too late in the paper. This summary of Open-OTAF, its expected strengths, and limitations, should have been discussed earlier in the paper to develop a clear intuition.

Weaknesses:

There are several clarity issues in the text that needs to be addressed. I was confused about important details in the paper, even after taking multiple passes through it. I am listing my major concerns here:

1. It is not clear if the paper is restricted to environments with a shared team reward or not. The description in Section 3 mentions that each agent has its own reward function (reward function being parameterized by the agent index), while the Figure 1 mentions that the reward is shared.

2. The term Marginal Utility is very confusing. This function has not been formally defined anywhere and is suddenly used at the end of Page 4 by using a relation with the Q function. A formal definition is required for this term, seeing how it is central to the paper.

3. Similarly, the term "teamwork situation" is very confusing to me. As example illustrating this is required. Also, the associated assumption in Page 4, "we assume that they may lead to similar teamwork situations at certain moments", needs justification in the paper.

4. There are several design choices introduced in Section 4 that are quite hard to follow. A clear intuition for each of these design choices along with descriptions of what they are expected to achieve needs to be presented. I particularly found the cluster model descriptions lacking in clarity. Firstly, how is it assumed that trajectories of past behaviours are a good proxy for current agent strategies (since clustering is done based on these trajectories). I would assume that past behaviours, especially ones that are several time steps into the past, are quite irrelevant to the current strategies of agents. Secondly, if the environment is partially observable, how are these trajectories for all agents obtained in the first place. I would assume that it is quite common to have situations where agents move in and out of view, and their action histories cannot be accurately observed by another representative agent. Thirdly, how are the number of clusters determined and misclassification errors handled. The experiments use 10 clusters, but this seems like an arbitrary choice. In the experiments, a study on the effect of varying the number of clusters needs to be performed.

5. Several results in the experiments are close and a statistical significance test is required. For example, in Table 2, the performance of Open-OTAF seems quite close to the performance of VAE_GPL.

6. The citations in the entire text need to be fixed. Citations are used as nouns in sentences where they are not supposed to be nouns (see \citet and \citep commands in latex and use them appropriately).


Minor Comments:

Pg 2: briefly review -> brief review

Pg 6: Then, we can then calculate -> We can then calculate

Pg 11: controlled agent adapts to -> controlled agent adapt to

---

> ### Author Response · Authors · 2025-06-06
> **Response to Reviewer nBpW**
>
> Thank you for your constructive feedback on our manuscript. We have modified the manuscript accordingly, and all the changes to your comments are highlighted in cyan in the revised manuscript.
>
> **Regarding the reward:** Thanks for pointing out this issue. Our framework supports both shared and individual reward settings, with the specific configuration determined by the problem domain. The general formulation (Section 3) employs parameterized reward functions $R^i$ indexed by agent $i$, enabling individualized objectives when necessary (e.g., role-dependent priorities). Figure 1 simplifies this to a shared reward R for schematic clarity, reflecting common collaborative scenarios where team-wide optimization is paramount.
>
> **Regarding the Marginal Utility:** Thanks for pointing out this issue. Our modifications are summarized as follows.
>
> 1) **Page 4/ Section 3, Paragraph 3:** We added the following statement regarding this issue:`` Traditional MARL often optimizes team utility directly via Q value function, but this fails to attribute individual contributions in ad hoc teams Gu et al. (2021)."
>
> 2) **Page 8/ Section 4.3, Paragraph 4:** We added the following statement regarding this issue:``We use a marginal utility network to estimate the ad hoc agent's marginal utility as $\hat{u}_t^i(\tau_t^i,a_t^i; C_t^i)\approx u^i(s_t, a_t^i,\textbf{a}_t^{-i})$, which representing its incremental contribution to the team’s expected return, conditioned on the current teamwork context. This quantifies how much $a^i$ improves team performance, given teammates' behaviors encoded in $C_t^i$."
>
> **Regarding the Teamwork Situation:** Thanks for pointing out this issue. In the revised manuscript, following your suggestions, we provided more details about the teamwork situation. Our modifications are summarized as follows.
>
> 1) **Page 5/ Section 3, Paragraph 4:** We defined the teamwork situation $C_t^i$ with an example. We added the following statement regarding this issue:`` To enable adaptive decision-making, the ad hoc agent’s policy must respond to teammate behaviors. We address the complexity of unknown teammates by encoding their interactions into a teamwork situation—a
> compact latent representation of the team’s strategic state. This representation captures how teammates collectively influence environmental dynamics, allowing the agent to optimize its behavior for the current cooperative context. For example, in our truck-drone delivery system, the teamwork situation embedding encodes the state of task phases, interaction patterns and so on."
>
> 2) **Page 5/ Section 3, Paragraph 5:** We added the following statement regarding this issue:`` Though different teammates generate diverse state-action trajectories, they may lead to similar teamwork situations at certain moments, and the ad hoc agent’s action would affect their transitions (Gu et al. (2021))."

---

> > ### Author Response · Authors · 2025-06-06
> > **Response to Reviewer nBpW (Cont.)**
> >
> > **Regarding Comment 4:** Thanks for pointing out the concerns.
> >
> > 1) Our key insight is that teamwork performance is jointly affected by the autonomous agent and other teammates’ behaviors. So, the agent’s optimal behavior depends on the current teamwork situation, which indicates the influence on the environmental dynamics caused by other teammates. In the global inference model, we use the state and teammates' actions to estimate the teamwork situation, which captures the core knowledge about other teammates’ current behaviors. The assumption is grounded in two principles: 1) While individual actions may vary, teamwork patterns often exhibit temporal consistency. Clustering past trajectories captures these recurrent high-level strategies, which remain relevant for inferring the current teamwork situation. 2) The ``teamwork situation" is modeled as a latent state aggregating historical interactions, not raw past actions. Thus, even outdated actions contribute to inferring the persistent teamwork context, which shapes current agent strategies. We acknowledge that highly non-stationary environments may reduce the relevance of distant past behaviors. To address this, our model weights recent trajectories more heavily with a discount factor $\gamma$ as shown in Equation 4 in the revised version.
> >
> >
> > 2) To overcome partial observability, we conduct the CTDE framework, where our method is granted access to the global state and other teammates' actions during training. During execution, we propose an information-based proxy encoder to implicitly infer the learned variables from local observations. Then, the autonomous agent adapts to new teammates' behaviors dynamically and quickly by conditioning its policy on the inferred variables. Specifically, we introduce a proxy encoder to estimate the teamwork situation from the local transition data $b_t^i$. We assume that $b_t^i$ can partly reflect the current teamwork situation since the transition implicitly indicates the underlying dynamics, which is primarily influenced by other teammates' behaviors.  Then, an information-based loss function $L_{MI}$ is used (as shown in Equation (6)) to maximize the conditional mutual information between the proxy variables $Z^i$ and the true variables $C^i$. Please refer to Page 9, Section 4.4 for more details.
> >
> > 3) We conduct new experiments in Section 5/Appendix to validate cluster sensitivity. Please refer to the **General Response of Experiment Result**.
> >
> >
> > **Regarding the statistical significance test:** As shown in Table 1 (in the revised version), we report the mean returns and 95% confidence intervals across 5 independent testing seeds, evaluated at the training checkpoint with peak validation performance. Our method, Open-OTAF, achieves statistically significant superiority over baselines (bolded values), with performance gains ranging from 5.1% to 49.8% in diverse teammate-type settings. This aligns with the expected correlation between training efficacy and generalization capability—methods demonstrating lower returns during training consistently underperformed when tested with novel teammates. While these results robustly demonstrate Open-OTAF's adaptability, future work will include formal significance testing to further validate these findings.

---

> > > ### Comment · Reviewer_nBpW · 2025-06-19
> > > **Reply to the response.**
> > >
> > > Thank you for the detailed response and changes to the manuscript. I am satisfied with all the responses. I am happy to recommend acceptance.

---

### Author Response · Authors · 2025-06-17
**Sincere Gratitude for Your Time and Constructive Feedback**

Dear Editor-in-Chief, Action Editors, and Reviewers,

I sincerely appreciate the opportunity to submit our work, "Collaboration with Dynamic Open Ad Hoc Teams via Team State Modelling," to Transactions on Machine Learning Research (TMLR). We are deeply grateful for your time, expertise, and the thoughtful feedback you provided, which has greatly enhanced the clarity of our manuscript.

The reviewers’ insightful comments were invaluable in refining key aspects of our work, particularly in improving the consistency of notations, clarifying technical details, and strengthening the experimental evaluation. We have carefully addressed all suggestions in the revised manuscript and believe these revisions have significantly elevated the contribution and impact of our study.

Thank you once again for your time and constructive suggestions.

Best regards,

The authors

---

### Decision · Action_Editor_27w1 · 2025-06-18

**Recommendation:** Accept with minor revision

**Additional Comments:**

Address minor notation/phrasing errors (Reviewer 69MY’s Sec.5–13.1 points) pre-publication. Future work should test real-world domains.

**Audience:**

Yes

**Audience Explanation:**

Open ad hoc teamwork addresses a critical gap in MARL for real-world applications (e.g., robotics, logistics) where agents collaborate with unknown, dynamic teammates.

**Claims And Evidence:**

Yes

**Claims Explanation:**

The authors demonstrate Open-OTAF's superiority through rigorous experiments across four benchmarks (LBF, Wolfpack, PCN, FortAttack), showing performance gains over baselines (Table 1). Reviewers initially questioned environmental simplicity and notation clarity, but authors justified benchmark choices (established in prior work [R1,R2]) and overhauled notation (Table R.1/R.2).

---

> ### Author Response · Authors · 2025-06-27
> **Regarding the camera-ready version**
>
> Dear Editors In Chief, Action Editors,
>
> We hope this message finds you well. We are writing to follow up regarding the camera-ready version we submitted for Paper 4631, "Collaboration with Dynamic Open Ad Hoc Team via Team State Modelling." We sincerely appreciate the time and effort you’ve dedicated to reviewing our work. We have carefully responded to all the comments raised so far and made the necessary revisions in the camera-ready version.
>
> We have carefully addressed all the comments and incorporated the requested revisions into the final version. To ensure a smooth process, we would like to confirm whether any additional clarifications or adjustments are required from our side. Please do not hesitate to let us know if further information or modifications would be helpful.
>
> Thank you again for your valuable feedback and consideration. We greatly appreciate your guidance and look forward to hearing from you at your earliest convenience.
>
> Best regards,
>
> The authors